# Isoniazid preventive therapy completion between July-September 2019: A comparison across HIV differentiated service delivery models in Uganda

**Levicatus Mugenyi**[1,2]*, **Proscovia Mukonzo Namuwenge**[3], **Simple Ouma**[1,4],
**Baker Bakashaba**[1], **Mastula Nanfuka**[1], **Jennifer Zech**[5], **Collins Agaba**[1], **Andrew Mijumbi
Ojok**[1], **Fedress Kaliba**[1], **John Bossa Kato**[1], **Ronald Opito**[1,6], **Yunus Miya**[1],
**Cordelia Katureebe**[3], **Yael Hirsch-Moverman**[5,7]

1 The AIDS Support Organization, Kampala, Uganda, 2 MRC/UVRI and LSHTM Uganda Research Unit,
Entebbe, Uganda, 3 Ministry of Health, Kampala, Uganda, 4 Institute of Clinical Trials and Methodology,
University College London, London, United Kingdom, 5 ICAP at Columbia University, New York, NY, United
States of America, 6 Department of Pubic Health, School of Health Sciences, Soroti University, Soroti,
Uganda, 7 Epidemiology Department, Mailman School of Public Health, Columbia University, New York, NY,
United States of America

* lmugenyi005@gmail.com

KENYA

**Data Availability Statement:** All relevant data are
within the paper and its Supporting Information
files.

## Abstract

### Background

Tuberculosis (TB) remains the leading cause of death among people living with HIV
(PLHIV). To prevent TB among PLHIV, the Ugandan national guidelines recommend Isonia-
zid Preventive Therapy (IPT) across differentiated service delivery (DSD) models, an effec-
tive way of delivering ART. DSD models include Community Drug Distribution Point
(CDDP), Community Client-led ART Delivery (CCLAD), Facility-Based Individual Manage-
ment (FBIM), Facility-Based Group (FBG), and Fast Track Drug Refill (FTDR). Little is
known about the impact of delivering IPT through DSD.

### Methods

We reviewed medical records of PLHIV who initiated IPT between June-September 2019 at
TASO Soroti (TS), Katakwi Hospital (KH) and Soroti Regional Referral Hospital (SRRH).
We defined IPT completion as completing a course of isoniazid within 6–9 months. We uti-
lized a modified Poisson regression to compare IPT completion across DSD models and
determine factors associated with IPT completion in each DSD model.

### Results

Data from 2968 PLHIV were reviewed (SRRH: 50.2%, TS: 25.8%, KH: 24.0%); females:
60.7%; first-line ART: 91.7%; and Integrase Strand Transfer Inhibitor (INSTI)-based regi-
men: 61.9%. At IPT initiation, the median age and duration on ART were 41.5 (interquartile
range [IQR]; 32.3–50.2) and 6.0 (IQR: 3.7–8.6) years, respectively. IPT completion overall

**Funding:** This project was funded by the Ugandan Ministry of Health and by grant # OPP1152764 from the Bill & Melinda Gates Foundation. The funders had no role in study design, data collection and analysis, decision to publish, or preparation of the manuscript.

**Competing interests:** The authors have declared that no competing interests exist

**Abbreviations:** DSD, Differentiated Services Delivery (model); CCLAD, Community Client-Led ART Delivery (model); CDDP, Community Drug Distribution Point (model); FTDR, Fast Track Drug Refill (model); FBG, Facility Based Group (model); TPT, TB Preventive Therapy; HIV, Human Immunodeficiency Virus; IQR, Interquartile range.

was 92.8% (95%CI: 91.8–93.7%); highest in CDDP (98.1%, 95%CI: 95.0–99.3%) and lowest in FBG (85.8%, 95%CI: 79.0–90.7%). Compared to FBIM, IPT completion was significantly higher in CDDP (adjusted rate ratio [aRR] = 1.15, 95%CI: 1.09–1.22) and CCLAD (aRR = 1.09, 95% CI 1.02–1.16). In facility-based models, IPT completion differed between sites (p<0.001). IPT completion increased with age for FBIM and CCLAD and was lower among female participants in the CCLAD (aRR = 0.82, 95%CI 0.67–0.97).

## Conclusion

IPT completion was high overall but highest in community-based models. Our findings provide evidence that supports integration of IPT within DSD models for ART delivery in Uganda and similar settings.

## Introduction

In 2020, an estimated 10 million people had tuberculosis (TB) disease, including one-quarter from sub-Saharan Africa [1]. Despite the impressive scale-up of HIV treatment, with 61.5% (23.3/37.9 million) people living with HIV (PLHIV) receiving antiretroviral therapy (ART) at the end of 2018, TB remains the leading cause of death among people living with HIV (PLHIV), accounting for approximately one-in-three AIDS-related deaths [2]. A quarter of the world's population is estimated to have latent TB infection (LTBI) [3]. PLHIV with LTBI are 20 times more likely to develop active TB disease within five years of the initial infection compared to people who are HIV-negative [4].

Prevention of TB disease using TB preventive therapy (TPT) is a critical component of the World Health Organization's (WHO) End TB Strategy [5]. Isoniazid preventive therapy (IPT), which involves daily administration of isoniazid and vitamin B6 for six months, was the most common TPT in resource-limited settings at the time of this study. In PLHIV and children aged <15 years (with or without HIV), IPT reduces the risk of developing TB by approximately 55% [6] and TB/HIV deaths by about 40% [7].

Emerging evidence suggests that IPT adherence and satisfaction can be improved by integrating TB and HIV services [8]. In a study from Botswana, men and younger clients cited work commitments as a cause of suboptimal IPT adherence [9]. It was noted from the Botswana study that targeted health outreach to the patient groups and expansion of locations or hours for accessing IPT (and ART) could improve completion [9]. In another study from Zimbabwe, IPT and ART refills were aligned, and clients self-selected their preferred models of care. The results indicated high IPT completion for both facility- and community-based models [10]. In Uganda, as the number of people initiated on ART in Uganda increased, differentiated service delivery (DSD) models that involve multi-month dispensing of ART and less frequent health facility visits (i.e., every 3–6 months) were adopted [11] as described in Fig 1. A 2011 study in Uganda found a significant difference in IPT completion between PLHIV receiving ART from a DSD model (72%) versus standard of care (53%) [12]. Health beliefs, social support, and perceived side-effects of IPT were some of the factors found to underlie the association between models of care and IPT completion [12]. Over time, the Ugandan Ministry of Health and development partners have put more resources into scale-up and differentiating IPT services [13, 14], yet only little is known about the impact of such efforts on IPT completion, especially across DSD models.

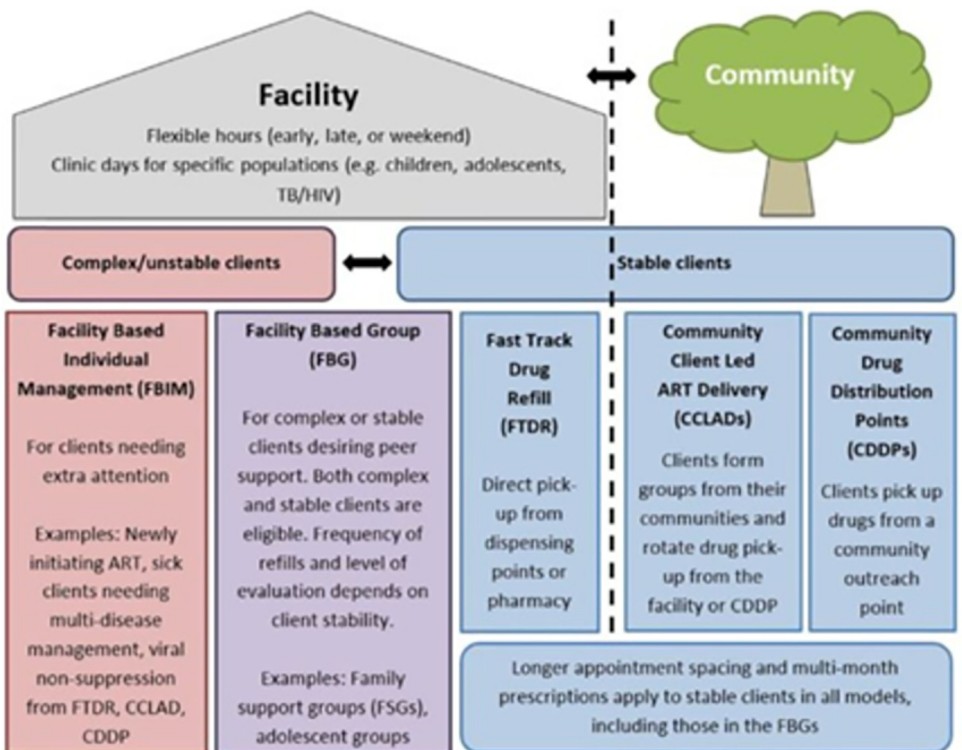

**Fig 1. An implementation guide for DSD models of HIV and TB services in Uganda.**

In this paper, we explored how IPT completion varied between Uganda's DSD models as well as factors associated with IPT completion in each DSD model. Findings will provide the much-needed evidence to guide future evidence-based policy and practice on the integration of TPT within DSD models for ART delivery, enhance the uptake and completion rate of TPT, and thus improve outcomes for PLHIV in Uganda and other similar resource-limited settings.

## Materials and methods

### Study design, setting, and population

This was a retrospective study with review of electronic medical records and patient registers for PLHIV who initiated IPT during the period of July-September 2019. The study was implemented in the Soroti region in Eastern Uganda which is approximately 290 km from Kampala. Soroti was selected because it has high-volume health facilities implementing IPT/ART integration across different DSD models. We purposively selected three high-volume health facilities that had been implementing IPT/ART integration for at least one year, including TASO Soroti, Katakwi hospital and Soroti regional referral hospital.

Uganda's DSD models (Fig 1) are either facility-based or community-based. Facility-based models include Facility Based Individual Management (FBIM), Facility Based Group (FBG), and Fast Track Drug Refill (FTDR); community-based models include Community Client Led ART Delivery (CCLAD) and Community Drug Distribution Points (CDDP). Stable clients, who are virally suppressed are allocated to CCLAD, CDDP and FTDR while unstable clients (not virally suppressed) are allocated to FBIM and FBG. The 2018 national Ugandan IPT guidelines recommended that all eligible PLHIV receive IPT monthly at health facilities for the duration of treatment to enable healthcare providers to monitor adherence, adverse events,

and TB symptoms. This created a discordance between monthly IPT refills and multi-month dispensing for ART and posed a challenge for IPT uptake and completion due to time and cost constraints associated with monthly visits to health facilities. Therefore, it became important to align IPT refills with multi-month dispensing of ART for stable clients.

Individuals who initiated IPT at the three selected health facilities during the study period were eligible for chart/data review. To be eligible for IPT, a client must be living with HIV and be on ART for any duration. Those who had active pulmonary TB disease were not eligible for IPT initiation. The number of clinic follow-up visits and months of IPT dispensed at each visit varied by DSD model. Clinically stable clients in the community-based DSD models received multi-month (3–6 months) of synchronized IPT and ART refills per visit, while clinically unstable clients in the facility-based DSD models received 1–2 months of synchronized IPT and ART refills per visit. To complete TPT, clients in the community-based DSD models required only 1–2 clinic visits, while those in the facility-based models required 3–6 clinic visits within the 6–9 months of IPT initiation.

## Sample size estimation

We conducted a medical records review of all PLHIV who had initiated IPT at the three selected health facilities during July-September 2019.

## Data collection

We extracted de-identified data from electronic data systems (e.g., UgandaEMR) at the health facilities. Where data were missing from the electronic data system, we cross-checked with patient care cards or IPT registers and abstracted data with the help of a digital data abstraction tool installed on tablet computers. Data abstracted included dates of IPT and ART initiation, ART regimen, DSD model, last observed/refill date of IPT, IPT completion status, date stopped IPT and reasons for stopping IPT, IPT side effects and dates reported, other chronic illnesses, and demographic characteristics. Data were extracted from March-May 2021. All those with missing documentation on ART initiation (48/3016; 1.6%) were excluded.

## Data analysis

Participants' baseline characteristics were summarized overall and stratified by DSD model using proportions (for categorical variables) and means with standard deviations (SD) or medians with interquartile ranges (IQR) as appropriate (for continuous variables). IPT completion was measured as a proportion of participants completing six months of IPT within 6–9 months of those who initiated IPT. IPT completion was calculated for each DSD model and compared across the models using the chi-square test. A modified Poisson regression model with robust standard errors was fitted to the IPT completion data, hence estimating rate ratios (RR) with 95% confidence intervals (CI) [15]. First, the Poisson model was fitted to estimate RR by comparing the DSD models adjusted for study site, gender, age at IPT initiation, ART regimen, ART duration, and ART treatment line. Later, the Poisson model was fitted for each DSD model to determine factors associated with IPT completion. To account for missing data, multiple imputation chained equations were used to generate 100 imputations. The Poisson model results were presented as unadjusted (accounting for no other variables), and as adjusted (accounting for other variables) showing results both with complete case analysis and multiple imputation. All variables with bi-variable p-value <0.1 were subjected to a multiple modified Poisson model and results presented as adjusted estimates. Variance inflation factor (VIF) was used to test for multicollinearity in independent variables. Associations with p-

values <0.05 were considered statistically significant. Analysis was conducted using Stata version 15.

## Ethics approval

The study was registered by the Uganda National Council for Science & Technology (HS1228ES). A waiver to review and extract data from existing medical records was obtained from the TASO-Research Ethics Committee (TASOREC/004/2020-UG-REC-009) and the Columbia University Irving Medical Center IRB (IRB-AAAS9111). Furthermore, we obtained administrative clearance from the Ugandan Ministry of Health, the district health officers, the executive director of TASO, and those in charge of the participating facilities. Data anonymization was implemented prior to data extraction.

## Results

### Characteristics of study participants by DSD model

Data were extracted on 2968 PLHIV who initiated IPT during the study period and whose data on ART initiation was available (TASO Soroti: 766 [25.8%]; Soroti regional referral hospital: 1526 [50.2%]; Katakwi hospital: 712 [24.0%]); majority 1801 (60.7%) were female. Overall, 1368 (46.1%) of participants were enrolled in FTDR; 963 (32.4%) FBIM; 288 (9.7%) CCLAD; 208 (7.0%) CDDP; and 141 (4.8%) FBG. Among 49.5% (1470/2968) of participants with available data on age at IPT initiation, the median (IQR) age at IPT initiation was 41.5 (32.3–50.2) years with the largest proportion (27.6%, 406/1470) in the 40–49 years age group. The median (IQR) duration on ART at IPT initiation was 6.1 (3.7–8.6) years. The majority 1837 (62.2%) of participants were on an integrase strand transfer inhibitor (INSTI)-based regimen and a small proportion 260 (8.7%) were on protease inhibitors (PI)-based regimen. A vast majority 2718 (91.7%) of participants were on first-line ART regimens (Table 1).

### IPT completion across DSD models

Overall, 2754 participants completed IPT (92.8%, 95%CI: 91.8–93.7%). Reasons for not completing IPT included side effects (n = 14), poor adherence (n = 6), pill burden (n = 4), advised by a health worker (n = 1), and lost dose (n = 1). The reason for not completing IPT was missing for a large majority of those who never completed IPT (187/214, 87.4%). Fig 2 shows the number and proportion of PLHIV who completed IPT across the five DSD models. Completion statistically differed between the five DSD models (p<0.001) with the highest completion among those in the CDDP model (98.1%, 95% CI: 95.0–99.3%) and the lowest among those in the FBG model (85.8%, 95% CI: 79.0–90.7%).

### Factors associated with IPT completion among PLHIV

Table 2 shows RR estimates for IPT completion comparing the five DSD models. The VIF values for the predictors in the model were less than 4 indicating no collinearity. Completion differed significantly across the different DSD models with community-based models having a higher probability of IPT completion than facility-based models. IPT completion was significantly higher in CDDP (complete case analysis adjusted RR [aRR] = 1.15, 95% CI 1.09–1.22; p<0.001) and in CCLAD (complete case analysis aRR = 1.09, 95% CI 1.02–1.16; p = 0.014) than in FBIM. However, with multiple imputation the association with CCLAD disappeared (p = 0.246).

Table 3 shows factors associated with IPT completion in each DSD model. Again, there was no collinearity because for all models the VIF values for the predictors were less than 3. For

**Table 1. Characteristics of participants by DSD models.**

| Characteristics | Total | Facility-Based Individual Management (FBIM) | Facility-Based Group (FBG) | Fast Track Drug Refill (FTDR) | Community Drug Distribution Point (CDDP) | Community Client-led ART Delivery (CCLAD) |
|---|---|---|---|---|---|---|
| Number extracted | 2,968 | 963 | 141 | 1,368 | 288 | 208 |
| **Site, n (%)** | | | | | | |
| TASO Soroti | 766 (25.8) | 142 (14.8) | 47 (33.3) | 194 (14.2) | 223 (77.4) | 160 (76.9) |
| Soroti regional referral hospital | 1,490 (50.2) | 608 (63.1) | 73 (51.8) | 766 (56.0) | 0 (0.0) | 43 (20.7) |
| Katakwi hospital | 712 (24.0) | 213 (22.1) | 21 (14.9) | 408 (29.8) | 65 (22.6) | 5 (2.4) |
| **Sex, n (%)** | | | | | | |
| Female | 1801 (60.7) | 584 (60.6) | 91 (64.5) | 784 (57.3) | 202 (70.1) | 140 (67.3) |
| Male | 1167 (39.3) | 379 (39.4) | 50 (35.5) | 584 (42.7) | 86 (29.9) | 68 (32.7) |
| **Age in years at IPT initiation, n (%)** | | | | | | |
| <18 | 132 (4.5) | 51 (5.3) | 44 (31.2) | 18 (1.3) | 16 (5.6) | 3 (1.4) |
| 18–29 | 172 (5.8) | 84 (8.7) | 8 (5.7) | 67 (4.9) | 11 (3.8) | 2 (1.0) |
| 30–39 | 370 (12.5) | 122 (12.7) | 10 (7.1) | 172 (12.6) | 42 (14.6) | 24 (11.5) |
| 40–49 | 406 (13.7) | 51 (5.3) | 3 (2.1) | 193 (14.1) | 106 (36.8) | 53 (25.5) |
| 50–59 | 275 (9.3) | 29 (3.0) | 2 (1.4) | 101 (7.4) | 79 (24.7) | 64 (30.8) |
| 60+ | 115 (3.9) | 15 (1.6) | 0 (0.0) | 47 (3.4) | 34 (11.8) | 19 (9.1) |
| Missing | 1,498 (50.5) | 611 (63.5) | 74 (52.5) | 770 (56.3) | 0 (0.0) | 43 (20.7) |
| Median (IQR)[©] | 41.5 (32.3,50.2) | 33.5 (24.8,40.6) | 16.2 (12.1,25.7) | 41.9 (34.2,49.7) | 48.1 (40.2,54.5) | 50.1 (44.1,56.2) |
| **Duration on ART at IPT initiation (years), n (%)** | | | | | | |
| <5 | 660 (22.2) | 319 (33.1) | 28 (19.9) | 292 (21.4) | 11 (3.8) | 10 (4.8) |
| 5–9 | 1020 (34.4) | 351 (36.5) | 48 (34.0) | 565 (41.3) | 36 (12.5) | 20 (9.6) |
| 10–14 | 375 (12.6) | 112 (11.6) | 15 (10.6) | 227 (16.6) | 7 (2.4) | 14 (6.7) |
| 15+ | 55 (1.9) | 23 (2.4) | 1 (0.7) | 27 (2.0) | 0 (0.0) | 4 (1.9) |
| Missing | 858 (28.9) | 158 (16.4) | 49 (34.8) | 257 (18.8) | 234 (81.3) | 160 (76.9) |
| Median (IQR)[©] | 6.0 (3.7,8.6) | 5.6 (2.4,8.0) | 5.9 (3.8,8.1) | 6.3 (4.4,9.2) | 6.2 (4.9,7.8) | 7.8 (5.6,13.0) |
| **ART regimen[*], n (%)** | | | | | | |
| INSTI | 1837 (61.9) | 524 (54.4) | 80 (56.7) | 833 (60.9) | 217 (75.4) | 183 (88.0) |
| NNRTIs | 870 (29.3) | 317 (32.9) | 30 (21.3) | 446 (32.6) | 59 (20.5) | 18 (8.7) |
| PI | 260 (8.8) | 122 (12.7) | 31 (22.0) | 88 (6.4) | 12 (4.2) | 7 (3.4) |
| Missing | 1 (0.03) | 0 (0.0) | 0 (0.0) | 1 (0.07) | 0 (0.0) | 0 (0.0) |
| **ART treatment line, n (%)** | | | | | | |
| First line | 2718 (91.6) | 854 (88.7) | 117 (83.0) | 1275 (93.2) | 275 (95.5) | 197 (94.7) |
| Second, third or fourth line | 247 (8.3) | 108 (11.2) | 24 (17.0) | 93 (6.8) | 13 (4.5) | 9 (4.3) |
| Missing | 3 (0.1) | 1 (0.1) | 0 (0.0) | 0 (0.0) | 0 (0.0) | 2 (1.0) |

[©]*Excluding those with missing data*

[*]*INSTI = integrase strand transfer inhibitor, NNRTIs = non-nucleoside reverse transcriptase inhibitors, PI = protease inhibitor*

each DSD model, the results from complete case analysis are similar to those from multiple imputation, which implies that the former method did not produce biased estimates. IPT completion differed between sites in facility-based models (FBIM, FBG and FTDR) but not in community-based models (CCLAD and CDDP) both with complete case analysis and multiple imputation. Below, we present results from multiple imputation to report on all participants.

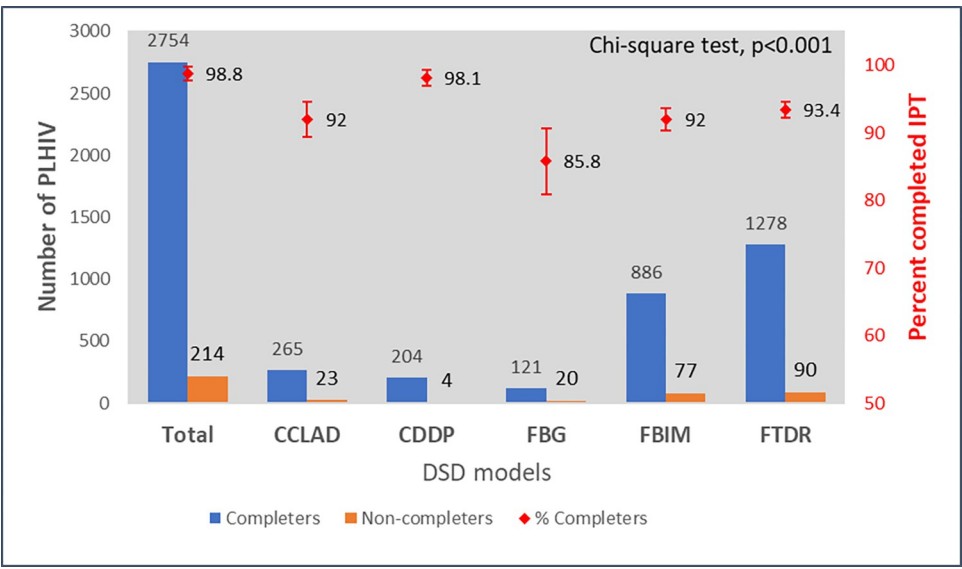

**Fig 2. Number and proportion of participants who completed IPT across DSD models.**

IPT completion within FBIM was significantly higher in Katakwi hospital than in TASO Soroti (aRR = 1.39, 95% CI 1.22–1.58) and in Soroti regional referral hospital than in TASO Soroti (aRR = 1.48, 95% CI 1.31–1.67). In addition, IPT completion within FBG was significantly higher in Katakwi hospital than in TASO Soroti (aRR = 1.53, 95% CI 1.20–1.97) and higher in Soroti regional referral hospital than TASO Soroti (aRR = 1.54, 95% CI 1.24–1.93). Further, IPT completion within FTDR was significantly higher in Katakwi hospital than in TASO Soroti (aRR = 1.26, 95% CI 1.16–1.37) and higher in Soroti regional referral hospital than in TASO Soroti (aRR = 1.29, 95% CI 1.19–1.40). Also, IPT completion increased with age at IPT initiation for FBIM and CCLAD, and it was higher among participants on PI-based than INSTI-based ART regimens only in CCLAD though not statistically significant after adjusting for other factors (aRR = 1.04, 95% CI 0.80–1.36). Lastly, IPT completion was lower among female than male participants in CCLAD (aRR = 0.82, 95% CI 0.67–0.97). None of the factors studied was associated with IPT completion among participants in CDDP.

## Discussion

We found a high rate of IPT completion (92.8%) in DSD models that aligned IPT and ART in the Ugandan setting. Generally, PLHIV who received care from community-based DSD

**Table 2. IPT completion rate ratio estimates comparing DSD models.**

| DSD model | Unadjusted estimates | | Adjusted estimates*: complete case analysis | | Adjusted estimates*: multiple imputation | |
|---|---|---|---|---|---|---|
| | RR (95% CI) | p | RR (95% CI) | p | RR (95% CI) | p |
| FBIM | 1.0 | | 1.0 | | 1.0 | |
| FBG | 0.93 (0.87, 1.00) | 0.050 | 0.99 (0.84, 1.17) | 0.898 | 0.99 (0.92, 1.07) | 0.787 |
| FTDR | 1.02 (0.99, 1.04) | 0.199 | 1.05 (0.98, 1.11) | 0.121 | 0.99 (0.97, 1.02) | 0.657 |
| CDDP | 1.07 (1.04, 1.09) | <0.001 | 1.15 (1.09, 1.22) | <0.001 | 1.03 (1.00, 1.06) | 0.031 |
| CCLAD | 1.00 (0.92, 1.04) | 0.996 | 1.09 (1.02, 1.16) | 0.014 | 0.98 (0.94, 1.02) | 0.246 |

*Adjusted for study site, gender, age at IPT initiation, ART duration, ART regimen and ART treatment line

**Table 3. Factors associated with IPT completion among PLHIV in each DSD model.**

| Factors© | Total | Completed n (%) | Unadjusted estimates | | Adjusted estimates: complete case analysis | | Adjusted estimates: multiple imputation | |
|---|---|---|---|---|---|---|---|---|
| | | | RR (95% CI) | P | RR (95% CI) | p | RR (95% CI) | p |
| **Facility-Based Individual Management (FBIM)** | | | | | | | | |
| **Site** | | | | | | | | |
| TASO Soroti | 142 | 93 (65.5) | 1.0 | | 1.0 | | 1.0 | |
| Katakwi hospital | 213 | 198 (93.0) | 1.42 (1.25–1.61) | <0.001 | 1.34 (1.15–1.57) | <0.001 | 1.39 (1.22–1.58) | <0.001 |
| Soroti regional referral hospital® | 608 | 595 (97.9) | 1.49 (1.33–1.68) | <0.001 | . . . | | 1.48 (1.31–1.67) | <0.001 |
| **Age at initiation of IPT in years** | | | | | | | | |
| <18 | 51 | 31 (60.8) | 1.0 | | 1.0 | | 1.0 | |
| 18–29 | 84 | 73 (86.9) | 1.43 (1.13–1.81) | 0.003 | 1.22 (0.93–1.60) | 0.147 | 1.08 (0.98–1.20) | 0.131 |
| 30–39 | 122 | 103 (84.4) | 1.39 (1.09–1.75) | 0.006 | 1.20 (0.92–1.57) | 0.176 | 1.08 (0.97–1.19) | 0.144 |
| 40–49 | 51 | 42 (82.4) | 1.35 (1.05–1.74) | 0.019 | 1.26 (0.96–1.65) | 0.093 | 1.10 (1.00–1.22) | 0.060 |
| 50–59 | 29 | 25 (86.2) | 1.42 (1.09–1.85) | 0.010 | 1.35 (1.03–1.77) | 0.030 | 1.12 (1.02–1.24) | 0.025 |
| 60+ | 15 | 14 (93.3) | 1.54 (1.19–1.99) | 0.001 | 1.52 (1.11–2.08) | 0.008 | 1.15 (1.03–2.29) | 0.010 |
| **ART regimen** | | | | | | | | |
| INSTI | 524 | 484 (92.4) | 1.00 | | 1.0 | | 1.0 | |
| NNRTIs | 317 | 297 (93.7) | 1.01 (0.98–1.05) | 0.460 | 1.09 (0.92–1.28) | 0.304 | 1.02 (0.98–1.07) | 0.327 |
| PI | 122 | 105 (86.1) | 0.93 (0.86–1.00) | 0.067 | 1.07 (0.87–1.32) | 0.536 | 1.01 (0.95–1.09) | 0.694 |
| **Facility-Based Group (FBG)** | | | | | | | | |
| **Site** | | | | | | | | |
| TASO Soroti (TS) | 47 | 29 (61.7) | 1.0 | | 1.0 | | 1.0 | |
| Katakwi Hospital (KH) | 21 | 20 (95.2) | 1.54 (1.21–1.97) | 0.001 | 1.54 (1.18–1.99) | 0.001 | 1.53 (1.20–1.97) | 0.001 |
| Soroti RRH (SRRH) ® | 73 | 72 (98.6) | 1.60 (1.27–2.01) | <0.001 | . . . | | 1.54 (1.24–1.93) | <0.001 |
| **Sex** | | | | | | | | |
| Male | 50 | 38 (76.0) | 1.0 | | 1.0 | | 1.0 | |
| Female | 91 | 83 (91.2) | 1.20 (1.01–1.42) | 0.034 | 1.24 (0.93–1.66) | 0.144 | 1.10 (0.95–1.28) | 0.209 |
| **Age at initiation of IPT in years¶** | | | | | | | | |
| <18 | 44 | 30 (68.2) | 1.0 | | 1.0 | | 1.0 | |
| 18–29 | 8 | 8 (100) | 1.47 (1.20–1.80) | <0.001 | 1.40 (1.11–1.77) | 0.005 | 1.11 (0.96–1.29) | 0.171 |
| 30–39 | 10 | 6 (60.0) | 0.88 (0.51–1.52) | 0.648 | 0.88 (0.55–1.39) | 0.576 | 0.96 (0.80–1.15) | 0.630 |
| 40+ | 5 | 4 (80.0) | 1.17 (0.72–1.91) | 0.519 | 1.06 (0.63–1.79) | 0.818 | 1.03 (0.83–1.28) | 0.761 |
| **Fast Track Drug Refill (FTDR)** | | | | | | | | |
| **Site** | | | | | | | | |
| TASO Soroti (TS) | 194 | 146 (75.3) | 1.0 | | 1.0 | | 1.0 | |
| Katakwi Hospital (KH) | 408 | 388 (95.1) | 1.26 (1.16–1.37) | <0.001 | 1.26 (1.16–1.37) | <0.001 | 1.26 (1.16–1.37) | <0.001 |
| Soroti RRH (SRRH) | 766 | 744 (97.1) | 1.29 (1.19–1.40) | <0.001 | 1.29 (1.19–1.40) | <0.001 | 1.29 (1.19–1.40) | <0.001 |
| **ART treatment line** | | | | | | | | |
| First line | 1275 | 1199 (94.0) | 1.0 | | 1.0 | | 1.0 | |
| Second, third or fourth line | 93 | 79 (85.0) | 0.90 (0.83–0.99) | 0.022 | 0.92 (0.85–1.00) | 0.053 | 0.92 (0.85–1.00) | 0.053 |
| **Community Client-led ART Delivery (CCLAD)** | | | | | | | | |
| **Sex** | | | | | | | | |
| Male | 86 | 83 (96.5) | 1.00 | | 1.0 | | 1.0 | |
| Female | 202 | 182 (90.1) | 0.93 (0.88–0.99) | 0.027 | 0.81 (0.69–0.97) | 0.020 | 0.82 (0.67–0.97) | 0.020 |
| **ART regimen** | | | | | | | | |
| INSTI | 217 | 202 (93.1) | 1.0 | | 1.0 | | 1.0 | |
| NNRTIs | 59 | 51 (86.4) | 0.93 (0.83–1.03) | 0.177 | 1.01 (0.84–1.21) | 0.914 | 1.01 (0.84–1.21) | 0.914 |
| PI | 12 | 12 (100) | 1.07 (1.04–1.11) | <0.001 | 1.04 (0.80–1.36) | 0.796 | 1.04 (0.80–1.35) | 0.796 |

*(Continued)*

**Table 3.** (Continued)

| Factors© | Total | Completed n (%) | Unadjusted estimates | | Adjusted estimates: complete case analysis | | Adjusted estimates: multiple imputation | |
|---|---|---|---|---|---|---|---|---|
| | | | RR (95% CI) | P | RR (95% CI) | p | RR (95% CI) | p |
| **Facility-Based Individual Management (FBIM)** | | | | | | | | |
| **Age at initiation of IPT in years¶** | | | | | | | | |
| <30 | 27 | 22 (81.5) | 1.0 | | 1.0 | | 1.0 | |
| 30–39 | 42 | 37 (88.1) | 1.08 (0.87–1.34) | 0.470 | 1.26 (0.54–2.94) | 0.585 | 1.26 (0.54–2.94) | 0.585 |
| 40–49 | 106 | 97 (91.5) | 1.12 (0.93–1.36) | 0.229 | 1.21 (0.52–2.81) | 0.658 | 1.21 (0.52–2.81) | 0.658 |
| 50–59 | 79 | 75 (94.9) | 1.17 (0.97–1.41) | 0.110 | 1.29 (0.57–2.95) | 0.545 | 1.29 (0.57–2.95) | 0.545 |
| 60+ | 34 | 34 (100) | 1.23 (1.02–1.47) | 0.026 | 1.43 (0.63–3.22) | 0.393 | 1.43 (0.63–3.22) | 0.393 |
| **Duration on ART at IPT initiation (years)¶** | | | | | | | | |
| <5 | 11 | 11 (100) | 1.0 | | 1.0 | | 1.0 | |
| 5–9 | 36 | 32 (89.0) | 0.89 (0.79–1.00) | 0.048 | 0.96 (0.85–1.07) | 0.423 | 0.96 (0.85–1.07) | 0.423 |
| 10+ | 7 | 5 (71.4) | 0.71 (0.45–1.15) | 0.163 | 0.72 (0.47–1.10) | 0.132 | 0.72 (0.47–1.10) | 0.132 |

©No factor was associated with IPT completion in the CDDP model; hence no results are shown for CDDP in this table; ®Data on age at IPT initiation was missing for all participants at Soroti regional referral hospital site, hence no estimates for this site with a model containing age; ¶Data regrouped due to small cell counts.

models (CDDP and CCLAD) had higher rates of IPT completion than their counterparts who were receiving care from facility-based DSD models (FBIM, FBG and FTDR). Further analysis revealed that unlike in the community-based models, IPT completion at facility-based models differed between sites. IPT completion significantly increased with increasing age at IPT initiation only for FBIM and FTDR, and it was lower among female participants within only the CCLAD model.

The high IPT completion rate we found in this study is similar to that reported by Sensalire *et al.* in 2019 of 89% among PLHIV attending 14 high-volume ART clinics in Uganda [16]. However, the completion rate found in this study is much higher than the one previously reported in 2016 from Eastern Uganda, a similar study setting, which reported IPT completion of 72% among PLHIV who received care from DSD models and 53% among those receiving the standard of care [12]. Likewise, another study during 2006–8 reported IPT completion of 33.6% among PLHIV in Uganda [12]. The low IPT completion rates reported in the earlier studies in Uganda can be attributed to limited campaigns and services for TB prevention during that time. Similarly, a study conducted in urban Zambia around the same time as our study found a high rate of IPT completion among PLHIV at 90.2% [17]. However, the Zambia study only looked at integrating IPT into the Fast Track (FT) model, whereas our study examined integration of IPT across multiple DSD models, including community- and facility-based models.

Unlike our study, two systematic reviews of latent tuberculosis cascade-of-care among PLHIV found very low completion rates as low as 33.2% [18] and 18.8% [19], which were much lower than we found in our study. The higher IPT completion found in our study compared to the above two systematic reviews is an indicator of the high quality of the IPT program in Uganda [20], which followed the implementation of IPT Surge activities by the Uganda Ministry of Health with technical and financial support from development partners. The IPT Surge activities were aimed at increasing IPT coverage, uptake, and completion across the country. PEPFAR launched the IPT Surge in 2018 through USAID and CDC projects that received increased funding for supplies and monitoring of IPT activities across the country [14]. Additionally, the Ministry of Health launched the 100-day IPT scale-up campaign on July

3, 2019, targeting all eligible PLHIV in Uganda [13, 14]. These campaigns, which were in line with the WHO target of increasing TPT coverage to at least 90% by the year 2025 [21], could have contributed to the high rate of IPT completion reported in this study. Furthermore, these campaigns were accompanied by a national quality improvement collaborative that also targeted IPT [16]. A quality improvement study in Nigeria showed marked improvement in IPT completion when barriers such as poor tracking systems and poor documentation of IPT were targeted [22].

Our finding of the higher IPT completion rate in community-based compared to facility-based DSD models is in line with a previous study in Uganda, which documented higher odds of IPT completion of 2.2 among PLHIV receiving differentiated care compared to those receiving non-differentiated care [23]. Studies in other African countries have suggested that community-based DSD models are acceptable for HIV service delivery [24, 25]. It is possible that the allocation of stable clients to community-based models could explain the higher rates of IPT completion among clients in these DSD models. Furthermore, IPT completion could have been improved among the community-based DSD models due to the reduction of travel and time costs resulting from picking drugs in the community [26]. The current finding showing a higher IPT completion rate among PLHIV in the community-based models than in the facility-based models supports the recommendation of alignment of prescribing practices for IPT with those for ART [8]. However, we think that a qualitative inquiry would help untangle these findings.

Predictors of IPT completion varied by the DSD model. IPT completion among facility-based DSD models (FBIM, FBG and FTDR) differed between sites, while the rate significantly increased with increasing age at IPT initiation for FBIM and CCLAD. There are only a few studies that explored factors affecting IPT completion in the setting of DSD models. One such study conducted in 2016 in five rural communities in Uganda also found that factors affecting IPT completion were significantly different across the models of care investigated [23]. Different from our study which looked at factors associated with IPT completion within each DSD model, the study conducted in 2016 in the rural communities of Uganda combined all DSD models and compared these to the standard of care. In addition, the 2016 study included only stable clients yet our study considers DSD models for both stable and unstable clients.

The results indicated that IPT completion was consistently lower in TASO Soroti facility models. Our investigations with the management of TASO Soroti facility revealed that around the study time, the facility experienced episodes of stock-out of IPT yet several clients were already initiated on it. This partly explains the comparatively lower IPT completion rate, especially among individuals in the facility-based models. By design, unlike those in the community-based models, PLHIV at facility-based models receive only one to two months of ART and IPT refills due to their being unstable on ART implying that unlike those in the community-based DSD models who required only 1–2 IPT refills to complete those doses, PLHIV in the facility-based DSD models would require up to 3–6 IPT refills to complete their doses. Therefore, during periods of frequent stock-out of IPT, facility-based clients were structurally disadvantaged and prone to poor TPT completion rate since several of them could not get IPT drug refills. Secondly, the poorer IPT completion rate at TASO Soroti could have been since PLHIV at the facility comprise of the most unstable groups with poorer adherence to ART.

IPT completion was lower among females than males in the CCLAD. This finding is contradictory to results from previous studies showing either no effect of sex on IPT completion among PLHIV [27–29] or males having poorer IPT completion rates than their female counterparts [30]. In addition, it's well known that males generally have poorer adherence to ART than their female counterparts [31]. Nevertheless, our study is unique in that we explored predictors of IPT completion by DSD models. Perhaps, there are some yet unknown reasons why

females in the CCLAD have poorer adherence to IPT compared to their male counterparts. Thus, there is a need for a follow-up study to better understand this phenomenon.

The increase in IPT completion associated with increasing age of participants reported in our paper concurs with the work by Little *et al*. (2019), who documented lower odds of IPT completion among participants aged <25 years compared to those aged >45 years (OR: 0.33, 95%CI: 0.18–0.60 [30]. The low rate of IPT completion among young people living with HIV could be attributed to poor adherence to HIV medication in this population [32–34].

The findings also showed that IPT completion was significantly higher in CDDP and CCLAD compared to the FBIM in a complete case analysis, however, with multiple imputation the association with CCLAD disappeared. This is likely due to the fact that none of the participants (0%) in CCLAD had missing data on age compared with more than 50% data on age missing for each of the other DSD models.

Our study had some limitations. First, the study was conducted in one region of the country and may not be generalizable to other regions in different settings. However, Soroti has high-volume health facilities implementing IPT/ART integration for the different DSD models. Second, there is potential for selection bias rooting from the fact that clients who are stable on ART and more likely to adhere to HIV treatment and TPT are selected into community DSD models while those who are unstable and less adherent to treatment receive care in facility-based models. Third, some clinical characteristics such as viral load and ART adherence were not extracted as information was not available, and some variables such as age had missing data. To prevent bias resulting from missing data, multiple imputation was performed, and results compared with those from a complete case analysis. Fourth, we could not verify IPT completion, and we relied on the medical record systems where the completion status was documented. The main strength of this study is that IPT completion was calculated based on a thorough medical record review of all PLHIV initiated on IPT in the study sites, which ensured increased precision and power to detect differences between and within DSD models.

In conclusion, this study revealed a high IPT completion rate; it was higher among PLHIV who received care from community-based DSD models than their counterparts in facility-based DSD models. Our findings provide evidence that supports further integration of IPT within DSD models for ART delivery in Uganda and other resource-limited settings.

## Supporting information

**S1 File. Inclusivity in global research.**
(DOCX)

**S1 Data.**
(ZIP)

## Acknowledgments

We extend our sincere thanks to the heads of the three study sites including SRRH, TS and KH. Special thanks go to the HIV patients whose data were used in this analysis. We thank all the research assistants who extracted these data.

## Author Contributions

**Conceptualization:** Levicatus Mugenyi, Proscovia Mukonzo Namuwenge, Baker Bakashaba, Mastula Nanfuka, Collins Agaba, Andrew Mijumbi Ojok, Cordelia Katureebe.

**Data curation:** Levicatus Mugenyi, Collins Agaba, Andrew Mijumbi Ojok, John Bossa Kato, Ronald Opito.

**Formal analysis:** Levicatus Mugenyi, Simple Ouma, Collins Agaba, John Bossa Kato, Cordelia Katureebe, Yael Hirsch-Moverman.

**Funding acquisition:** Baker Bakashaba, Jennifer Zech, Cordelia Katureebe, Yael Hirsch-Moverman.

**Investigation:** Levicatus Mugenyi, Proscovia Mukonzo Namuwenge, Baker Bakashaba, Collins Agaba, Andrew Mijumbi Ojok, Fedress Kaliba, John Bossa Kato, Ronald Opito, Cordelia Katureebe, Yael Hirsch-Moverman.

**Methodology:** Levicatus Mugenyi, Simple Ouma, Baker Bakashaba, Mastula Nanfuka, Collins Agaba, Ronald Opito, Cordelia Katureebe, Yael Hirsch-Moverman.

**Project administration:** Jennifer Zech, Fedress Kaliba, Ronald Opito, Yunus Miya.

**Software:** Levicatus Mugenyi.

**Supervision:** Levicatus Mugenyi, Proscovia Mukonzo Namuwenge, Collins Agaba, Ronald Opito, Yunus Miya, Cordelia Katureebe, Yael Hirsch-Moverman.

**Validation:** Levicatus Mugenyi, Proscovia Mukonzo Namuwenge, Simple Ouma, Baker Bakashaba, Cordelia Katureebe, Yael Hirsch-Moverman.

**Visualization:** Levicatus Mugenyi, Yael Hirsch-Moverman.

**Writing – original draft:** Levicatus Mugenyi, Simple Ouma.

**Writing – review & editing:** Levicatus Mugenyi, Proscovia Mukonzo Namuwenge, Simple Ouma, Baker Bakashaba, Mastula Nanfuka, Jennifer Zech, Collins Agaba, Andrew Mijumbi Ojok, Fedress Kaliba, John Bossa Kato, Ronald Opito, Yunus Miya, Cordelia Katureebe, Yael Hirsch-Moverman.

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
