## [Decision Letter · Decision Letter 0]

17 Aug 2022

PONE-D-22-21537Completion of Isoniazid Preventive Therapy among People Living with HIV on ART across Differentiated Service Delivery (DSD) Models in UgandaPLOS ONE

Dear Dr. Mugenyi,

Thank you for submitting your manuscript to PLOS ONE. After careful consideration, we feel that it has merit but does not fully meet PLOS ONE’s publication criteria as it currently stands. Therefore, we invite you to submit a revised version of the manuscript that addresses the points raised during the review process.--------------------------------------------------abstract: difficult to read due to full of abbreviations, defined as well as undefined.

introduction: excessive use of abbreviations makes difficult to read and follow.

methods:

multiple imputation using multivariate normal. How this was used for imputing categorical variables? Proportion of missing data is very large, over 50% sometimes. How good the imputation.what approach was used to identify factors associated with IPT when used modified Poisson model?Have authors did proper data cleaning prior to analysis? As consistent with the manuscript, there are 2968 people in submitted dataset. But among them only 2942 unique people identifiers. Have authors checked data for duplicates? One of the person in the dataset is 118.9 year old at the initiation of IPT, therefore more than 119 years old at the completion of IPT, Is this correct?-------------------------------------------------- Please submit your revised manuscript by Oct 01 2022 11:59PM. If you will need more time than this to complete your revisions, please reply to this message or contact the journal office at plosone@plos.org. Please include the following items when submitting your revised manuscript:A rebuttal letter that responds to each point raised by the academic editor and reviewer(s). You should upload this letter as a separate file labeled 'Response to Reviewers'.A marked-up copy of your manuscript that highlights changes made to the original version. You should upload this as a separate file labeled 'Revised Manuscript with Track Changes'.An unmarked version of your revised paper without tracked changes. You should upload this as a separate file labeled 'Manuscript'.

We look forward to receiving your revised manuscript.

Kind regards,

Ari Sam, PhD

Academic Editor

PLOS ONE

Journal Requirements:

3. In the ethics statement in the manuscript and in the online submission form, please provide additional information about the patient records/samples used in your retrospective study. Specifically, please ensure that you have discussed whether all data/samples were fully anonymized before you accessed them and/or whether the IRB or ethics committee waived the requirement for informed consent. If patients provided informed written consent to have data/samples from their medical records used in research, please include this information.

Reviewers' comments:

Reviewer's Responses to Questions

**Comments to the Author**

1. Is the manuscript technically sound, and do the data support the conclusions?

Reviewer #1: Yes

Reviewer #2: Partly

2. Has the statistical analysis been performed appropriately and rigorously? 

Reviewer #1: Yes

Reviewer #2: Yes

3. Have the authors made all data underlying the findings in their manuscript fully available?

Reviewer #1: Yes

Reviewer #2: Yes

4. Is the manuscript presented in an intelligible fashion and written in standard English?

Reviewer #1: Yes

Reviewer #2: Yes

5. Review Comments to the Author

Reviewer #1: Dear PLOSE One Team of editorials, thank you for the chance given to me to review “Completion of Isoniazid Preventive Therapy among People Living with HIV on ART across Differentiated Service Delivery (DSD) Models in Uganda”. The research article has its own importance for the improving the health of PLWH in particular and the general public in general. The following are my comments.

1. General Comments

The justification to conduct the study was not strong especially in the abstract section.

Why Isoniazid preventive therapy? Is PLWH are significant in number? Why across differentiated service? Why not on the particular service? IS that across the nations? IS the program new?

Use of correct words, sentences, grammar, paragraph and the whole sections content should be well revised.

Avoid use of abbreviations in the abstract section.

Use confidence interval whenever describing the percentage of the main objective.

Address the consistency from the beginning to the end of the article.

Add in to key words. E.g. Uganda.

2. Specific Comments

• Make sure that all the elements of the background section are fulfilled. Describe what is known and unknown and what gaps you want to fill by your study.

• In the result of the abstract section it is not mandatory to report both the OR and the P value since AOR is stronger than the P value.

• It is use of secondary data? Is the data complete? How Did you assess the validity and reliability of the study? Are the data sharing criteria met? Is that freely accessible or accessible per request? What are the steps taken hence?

• Did the contents of all sections have well ensured entailing all the contents? E.g. did the background, methods, results and discussions contains what it has intended to contain?

• Use correct tense, grammar, sentence, language, paraphrase, consistency…etc needs meticulous correction.

• Which type of data should be presented textually, by chart and by table?

• Revisit the numbers, totals and the statistics in general.

Reviewer #2: Thank you for your submission. Although the article presents a interesting finding, its applicability, originality and contribution to existing knowledge base is questionable. A recent publication from LMIC settings include:

Mukumbwa-Mwenechanya M, Mubiana M, Somwe P, Simwenda M, Zyongwe N, Kalumkumya E, Mwango L, Rabkin M, Mpasela F, Chungu F, Mwanza F. Integrating Isoniazid Preventive Therapy into the Fast-Track HIV Treatment Model in Urban Zambia: A Proof-of-Concept Pilot Project. medRxiv. 2022 Jan 1.

Guthrie T, Muheki C, Rosen S, Kanoowe S, Lagony S, Greener R, Miot J, Balidawa H, Kiggundu J, Calnan J, Dejene S. Similar costs and outcomes for differentiated service delivery models for HIV treatment in Uganda. medRxiv. 2021 Jan 1.

6. PLOS authors have the option to publish the peer review history of their article (what does this mean?). If published, this will include your full peer review and any attached files.

Reviewer #1: No

Reviewer #2: No

---

## [Author Response · Author response to Decision Letter 0]

10 Nov 2022

Response to reviewers 

Journal Requirements:

Response: Manuscript has been revised to address all PLOS ONE style requirements.

Response: A copy of PLOS questionnaire on inclusivity in global research has been attached as supporting information.

3. In the ethics statement in the manuscript and in the online submission form, please provide additional information about the patient records/samples used in your retrospective study. Specifically, please ensure that you have discussed whether all data/samples were fully anonymized before you accessed them and/or whether the IRB or ethics committee waived the requirement for informed consent. If patients provided informed written consent to have data/samples from their medical records used in research, please include this information.

Response: The ethics section has been revised as follows: “The study was registered by the Uganda National Council for Science & Technology (HS1228ES). A waiver to review and extract data from existing medical records was obtained from the TASO-Research Ethics Committee (TASOREC/004/2020-UG-REC-009) and the Columbia University Irving Medical Center IRB (IRB-AAAS9111). Furthermore, we obtained administrative clearance from the Ugandan Ministry of Health, the district health officers, the executive director of TASO, and the In-charges of the participating facilities. Data anonymization was implemented prior to data extraction.” 

Response: The ethics statement has been revised as requested and moved to the methods section.

Reviewers' comments:

Reviewer #1: 

Dear PLOS One Team of editorials, thank you for the chance given to me to review “Completion of Isoniazid Preventive Therapy among People Living with HIV on ART across Differentiated Service Delivery (DSD) Models in Uganda”. The research article has its own importance for the improving the health of PLWH in particular and the general public in general. The following are my comments.

Response: Thank you for acknowledging the importance of this research article.

1. General Comments

The justification to conduct the study was not strong especially in the abstract section.

Why Isoniazid preventive therapy? Is PLWH are significant in number? Why across differentiated service? Why not on the particular service? IS that across the nations? IS the program new?

Response: We have strengthened the rationale for this study in the Abstract (Lines 34-35) and the Introduction section (lines 66-78). The recommended regimen for TB prevention among PLHIV in Uganda is IPT delivered via DSD models, as those have shown to be effective in delivering ART to PLHIV. Because little is known about the impact of delivering IPT through different DSD models, we determined IPT completion and associated factors across DSD models.

Use of correct words, sentences, grammar, paragraph and the whole sections content should be well revised.

Response: Thanks for noting this. The authors conducted a careful review and revised the current manuscript for grammatical and spelling errors.

Avoid use of abbreviations in the abstract section.

Response: We minimized use of abbreviations to the extent possible and ensured that every abbreviation is defined on first use, with the exception of HIV which is in the English dictionary.

Use confidence interval whenever describing the percentage of the main objective.

Response: We updated the manuscript to include confidence intervals.

Address the consistency from the beginning to the end of the article.

Response: Consistency has been addressed in the revised manuscript.

Add in to key words. E.g. Uganda.

Response: Uganda has been added as a key word.

2. Specific Comments

• Make sure that all the elements of the background section are fulfilled. Describe what is known and unknown and what gaps you want to fill by your study.

Response: Thank you for pointing this out. We have revised the introduction and described what is known and unknown, and explicitly explained the gap that our study fills (lines 92-108). 

• In the result of the abstract section it is not mandatory to report both the OR and the P value since AOR is stronger than the P value.

Response: We agree with the reviewer and have removed p-values from the abstract where aRR is reported.

• It is use of secondary data? Is the data complete? How Did you assess the validity and reliability of the study? Are the data sharing criteria met? Is that freely accessible or accessible per request? What are the steps taken hence?

Response: Yes, the study used secondary data with some data missing on some variables. We used multiple imputation to handle missing data. All data used in the analysis has been made available as supplementary material. 

• Did the contents of all sections have well ensured entailing all the contents? E.g. did the background, methods, results and discussions contains what it has intended to contain?

Response: After making some adjustments, we confirm that all sections contain the intended relevant details. 

• Use correct tense, grammar, sentence, language, paraphrase, consistency…etc needs meticulous correction.

Response: Thank you for pointing this out. We have conducted a thorough review and corrected grammatical and spelling errors.

• Which type of data should be presented textually, by chart and by table?

Response: Because we were interested in stratifying our results by DSD models, we chose to present the results in tables with summary text.

• Revisit the numbers, totals and the statistics in general.

Response: We have conducted a thorough review to ensure that all numbers, totals, and statistics are accurate.

Reviewer #2: 

Thank you for your submission. Although the article presents a interesting finding, its applicability, originality and contribution to existing knowledge base is questionable. A recent publication from LMIC settings include:

Mukumbwa-Mwenechanya M, Mubiana M, Somwe P, Simwenda M, Zyongwe N, Kalumkumya E, Mwango L, Rabkin M, Mpasela F, Chungu F, Mwanza F. Integrating Isoniazid Preventive Therapy into the Fast-Track HIV Treatment Model in Urban Zambia: A Proof-of-Concept Pilot Project. medRxiv. 2022 Jan 1.

Guthrie T, Muheki C, Rosen S, Kanoowe S, Lagony S, Greener R, Miot J, Balidawa H, Kiggundu J, Calnan J, Dejene S. Similar costs and outcomes for differentiated service delivery models for HIV treatment in Uganda. medRxiv. 2021 Jan 1.

Response: Thank you for highlighting these two recent preprint publications. We believe that these studies do not dimmish from the applicability, originality and contribution of our article to the existing knowledge.

The study from Zambia, which was conducted around the same time as our study, is relevant and have added some of its findings to our Discussion section. However, please note that the Zambia study only looked at integrating IPT into the FT model, whereas our study examined integration of IPT across multiple DSD models, including community- and facility-based models. The study from Uganda mentioned above, measured cost per patient and cost per patient virally suppressed on ART in Uganda across five DSD models but did not consider the integration of IPT into these DSD models.

---

## [Editor Report · Decision Letter 1]

15 Nov 2022

PONE-D-22-21537R1Completion of IsoniazidPreventive Therapy among People Living with HIV on ART across Differentiated Service Delivery (DSD) Models in UgandaPLOS ONE

Dear Dr. Mugenyi,

Thank you for submitting your manuscript to PLOS ONE. After careful consideration, we feel that it has merit but does not fully meet PLOS ONE’s publication criteria as it currently stands. Therefore, we invite you to submit a revised version of the manuscript that addresses the points raised during the review process.

PLEASE SEE BELOW FOR SPECIFIC COMMENT(S)

We look forward to receiving your revised manuscript.

Kind regards,

Ari Samaranayaka, PhD

Academic Editor

PLOS ONE

Additional Editor Comments:

Dear authors, we noted some of the issues raised by the editor and reviewers on the original submission has not been addressed at the revised submission. Please make sure all of them were addressed.
---

## [Author Response · Author response to Decision Letter 1]

19 Nov 2022

Editor's comments sent on 18 November 2022.

1. Abstract: difficult to read due to full of abbreviations, defined as well as undefined.

Response: The abstract and the whole manuscript was revised to ensure that abbreviations are defined at their first mention before they can be used

2. Introduction: excessive use of abbreviations makes difficult to read and follow.

Response: All abbreviations are defined at their first mention in the introduction and throughout the manuscript.

3. methods:

• multiple imputation using multivariate normal. How this was used for imputing categorical variables? Proportion of missing data is very large, over 50% sometimes. How good the imputation.

Response: For multiple imputation using multivariate normal, dummy variables for all categorical variables were used. Yes, we acknowledge a large proportion of missing data for age (50.5%) which could have affected the imputed results. The following limitation has been added to the discussion “Second, missing data for some variables like age was large and this could have affected multiple imputation results on such variables” Line 278-279

• what approach was used to identify factors associated with IPT when used modified Poisson model?

Response: Since we were working with a few variables, all these were entered into a multiple modified Poisson model and backward model reduction considered by dropping factors with bigger p values. To address this the following statement has been included in the data analysis section “All variables were subjected to a multiple modified Poisson model and backward model building approach used by dropping the most statistically insignificant variables one at a time until a best fit was obtained.” Line 143-145

• Have authors did proper data cleaning prior to analysis? As consistent with the manuscript, there are 2968 people in submitted dataset. But among them only 2942 unique people identifiers. Have authors checked data for duplicates? One of the person in the dataset is 118.9 year old at the initiation of IPT, therefore more than 119 years old at the completion of IPT, Is this correct?

Response: Thank you very much for these observations and we agree with you regarding the observed issues. We confirm that data cleaning was conducted before data analysis. Regarding the duplicates, it is true there are some patients who were assigned same study IDs but the data in the other variables for these patients (except for two study IDs, KHC05519 and KHC06086) are unique and so these cases were different patients (except the two). Also, we noted the patient aged 118.9 years at IPT initiation. After reanalyzing the data excluding the duplicates for the two patients and the one patient aged 118.9 years, the results and conclusion remained consistent as reported in the manuscript. We therefore did not modify the results.

Earlier responses to reviewers

Response to reviewers 

Journal Requirements:

Response: Manuscript has been revised to address all PLOS ONE style requirements.

Response: A copy of PLOS questionnaire on inclusivity in global research has been attached as supporting information.

3. In the ethics statement in the manuscript and in the online submission form, please provide additional information about the patient records/samples used in your retrospective study. Specifically, please ensure that you have discussed whether all data/samples were fully anonymized before you accessed them and/or whether the IRB or ethics committee waived the requirement for informed consent. If patients provided informed written consent to have data/samples from their medical records used in research, please include this information.

Response: The ethics section has been revised as follows: “The study was registered by the Uganda National Council for Science & Technology (HS1228ES). A waiver to review and extract data from existing medical records was obtained from the TASO-Research Ethics Committee (TASOREC/004/2020-UG-REC-009) and the Columbia University Irving Medical Center IRB (IRB-AAAS9111). Furthermore, we obtained administrative clearance from the Ugandan Ministry of Health, the district health officers, the executive director of TASO, and the In-charges of the participating facilities. Data anonymization was implemented prior to data extraction.” 

Response: The ethics statement has been revised as requested and moved to the methods section.

Reviewers' comments:

Reviewer #1: 

Dear PLOS One Team of editorials, thank you for the chance given to me to review “Completion of Isoniazid Preventive Therapy among People Living with HIV on ART across Differentiated Service Delivery (DSD) Models in Uganda”. The research article has its own importance for the improving the health of PLWH in particular and the general public in general. The following are my comments.

Response: Thank you for acknowledging the importance of this research article.

1. General Comments

The justification to conduct the study was not strong especially in the abstract section.

Why Isoniazid preventive therapy? Is PLWH are significant in number? Why across differentiated service? Why not on the particular service? IS that across the nations? IS the program new?

Response: We have strengthened the rationale for this study in the Abstract (Lines 34-35) and the Introduction section (lines 66-78). The recommended regimen for TB prevention among PLHIV in Uganda is IPT delivered via DSD models, as those have shown to be effective in delivering ART to PLHIV. Because little is known about the impact of delivering IPT through different DSD models, we determined IPT completion and associated factors across DSD models.

Use of correct words, sentences, grammar, paragraph and the whole sections content should be well revised.

Response: Thanks for noting this. The authors conducted a careful review and revised the current manuscript for grammatical and spelling errors.

Avoid use of abbreviations in the abstract section.

Response: We minimized use of abbreviations to the extent possible and ensured that every abbreviation is defined on first use, with the exception of HIV which is in the English dictionary.

Use confidence interval whenever describing the percentage of the main objective.

Response: We updated the manuscript to include confidence intervals.

Address the consistency from the beginning to the end of the article.

Response: Consistency has been addressed in the revised manuscript.

Add in to key words. E.g. Uganda.

Response: Uganda has been added as a key word.

2. Specific Comments

• Make sure that all the elements of the background section are fulfilled. Describe what is known and unknown and what gaps you want to fill by your study.

Response: Thank you for pointing this out. We have revised the introduction and described what is known and unknown, and explicitly explained the gap that our study fills (lines 92-108). 

• In the result of the abstract section it is not mandatory to report both the OR and the P value since AOR is stronger than the P value.

Response: We agree with the reviewer and have removed p-values from the abstract where aRR is reported.

• It is use of secondary data? Is the data complete? How Did you assess the validity and reliability of the study? Are the data sharing criteria met? Is that freely accessible or accessible per request? What are the steps taken hence?

Response: Yes, the study used secondary data with some data missing on some variables. We used multiple imputation to handle missing data. All data used in the analysis has been made available as supplementary material. 

• Did the contents of all sections have well ensured entailing all the contents? E.g. did the background, methods, results and discussions contains what it has intended to contain?

Response: After making some adjustments, we confirm that all sections contain the intended relevant details. 

• Use correct tense, grammar, sentence, language, paraphrase, consistency…etc needs meticulous correction.

Response: Thank you for pointing this out. We have conducted a thorough review and corrected grammatical and spelling errors.

• Which type of data should be presented textually, by chart and by table?

Response: Because we were interested in stratifying our results by DSD models, we chose to present the results in tables with summary text.

• Revisit the numbers, totals and the statistics in general.

Response: We have conducted a thorough review to ensure that all numbers, totals, and statistics are accurate.

Reviewer #2: 

Thank you for your submission. Although the article presents a interesting finding, its applicability, originality and contribution to existing knowledge base is questionable. A recent publication from LMIC settings include:

Mukumbwa-Mwenechanya M, Mubiana M, Somwe P, Simwenda M, Zyongwe N, Kalumkumya E, Mwango L, Rabkin M, Mpasela F, Chungu F, Mwanza F. Integrating Isoniazid Preventive Therapy into the Fast-Track HIV Treatment Model in Urban Zambia: A Proof-of-Concept Pilot Project. medRxiv. 2022 Jan 1.

Guthrie T, Muheki C, Rosen S, Kanoowe S, Lagony S, Greener R, Miot J, Balidawa H, Kiggundu J, Calnan J, Dejene S. Similar costs and outcomes for differentiated service delivery models for HIV treatment in Uganda. medRxiv. 2021 Jan 1.

Response: Thank you for highlighting these two recent preprint publications. We believe that these studies do not dimmish from the applicability, originality and contribution of our article to the existing knowledge.

The study from Zambia, which was conducted around the same time as our study, is relevant and have added some of its findings to our Discussion section. However, please note that the Zambia study only looked at integrating IPT into the FT model, whereas our study examined integration of IPT across multiple DSD models, including community- and facility-based models. The study from Uganda mentioned above, measured cost per patient and cost per patient virally suppressed on ART in Uganda across five DSD models but did not consider the integration of IPT into these DSD models.

---

## [Decision Letter · Decision Letter 2]

7 Dec 2022

PONE-D-22-21537R2Completion of IsoniazidPreventive Therapy among People Living with HIV on ART across Differentiated Service Delivery (DSD) Models in UgandaPLOS ONE

Dear Dr. Mugenyi,

Thank you for submitting your manuscript to PLOS ONE. After careful consideration, we feel that it has merit but does not fully meet PLOS ONE’s publication criteria as it currently stands. Therefore, we invite you to submit a revised version of the manuscript that addresses the points raised during the review process.

We look forward to receiving your revised manuscript.

Kind regards,

Ari Samaranayaka, PhD

Academic Editor

PLOS ONE

Additional Editor Comments (if provided):

In the previous review I questioned the multiple imputation for categorical variables using multivariate normal. Authors responded saying it was done using dummy variables for all categorical variables. I believe this is impossible to do. Because, all data generated by multivariate normal process for missing values are in continuous scales, not in dummy coded scales as required by categorical variables. Please include a clarification for this within the manuscript with a suitable reference.

Reviewers' comments:

Reviewer's Responses to Questions

**Comments to the Author**

1. If the authors have adequately addressed your comments raised in a previous round of review and you feel that this manuscript is now acceptable for publication, you may indicate that here to bypass the “Comments to the Author” section, enter your conflict of interest statement in the “Confidential to Editor” section, and submit your "Accept" recommendation.

Reviewer #1: All comments have been addressed

Reviewer #2: All comments have been addressed

2. Is the manuscript technically sound, and do the data support the conclusions?

Reviewer #1: No

Reviewer #2: Yes

3. Has the statistical analysis been performed appropriately and rigorously? 

Reviewer #1: Yes

Reviewer #2: Yes

4. Have the authors made all data underlying the findings in their manuscript fully available?

Reviewer #1: Yes

Reviewer #2: Yes

5. Is the manuscript presented in an intelligible fashion and written in standard English?

Reviewer #1: Yes

Reviewer #2: Yes

6. Review Comments to the Author

Reviewer #1: Dear PLOS ONE team of editorials, I am thankful for the chance given to me to review a manuscript titled “Completion of Isoniazid Preventive Therapy among People Living with HIV on ART across Differentiated Service Delivery (DSD) Models in Uganda”. It gives new approach of reaching the PLHIV to prevent tuberculosis and other bacterial infections. However, you have reported low magnitude of Isoniazid prophylaxis by the DSD mode which is expected initially since you have PLHIV who are using it via other methods. The following are my comments;

a. The title should be changed in a way that best suit it and add time period.

b. It is comparison between the (a) the old and (b) new model of Isoniazid delivery. Hence, the method should be comparative study design. The study had assessed the status of completion of the Isoniazid prophylaxis and the findings suggest that the old model had better completion due to known reasons. Hence, what is new? It is an implementation study and what type of implementation study is that?

c. You didn’t include the clinical characteristics like viral load since its completion study. What about the status of completeness of data, data quality and how you had treated the data.

d. It is senses and you have conducted regression analysis which is Poisson. Is that the recommended solution? If it is even survey, is the case to variable ratio fits the model?

e. All the sections of the manuscript should entail what scientifically it intends to entail.

f. The Manuscript should be geared with the format of the PLOS ONE and language and statistics should be revisited and appropriate analysis and assumptions should be used and reported.

Reviewer #2: The authors have addressed the points which were mentioned in the previous review. The responses are satisfactory and clarify the points previously highlighted.

7. PLOS authors have the option to publish the peer review history of their article (what does this mean?). If published, this will include your full peer review and any attached files.

Reviewer #1: No

Reviewer #2: No

---

## [Author Response · Author response to Decision Letter 2]

6 Jan 2023

We thank the Reviewers for their thoughtful comments regarding our manuscript and the opportunity to revise our manuscript. We have revised the manuscript, which we believe has been substantially improved by incorporating the changes recommended by the Reviewers. Below, we provide a point-by-point response to the Reviewers’ comments and reference corresponding line numbers in the text with these changes. 

a. The title should be changed in a way that best suit it and add a time period. 

Response: Thank you for this excellent suggestion. The title has been revised and now reads as follows “Isoniazid Preventive Therapy Completion between July-September 2019: A Comparison across HIV Differentiated Service Delivery Models in Uganda”

b. It is comparison between the (a) the old and (b) new model of Isoniazid delivery. Hence, the method should be comparative study design. The study had assessed the status of completion of the Isoniazid prophylaxis and the findings suggest that the old model had better completion due to known reasons. Hence, what is new? It is an implementation study and what type of implementation study is that? 

Response: Dear reviewer, thank you for this suggestion. While we are citing past known IPT completion rates from other national explorations of old models of IPT delivery, we are not directly conducting a pre and post comparison of IPT completion rates. In this paper, we explore how IPT completion differed across various approaches of IPT delivery, known as differentiated service delivery (DSD) models. We updated the title to clarify that.

DSD models in Uganda have been implemented over the last 4-11 years depending on the implementer. However, other services like IPT were not layered into them but were left for facility-based individual models. The government eventually layered IPT within the different DSD models and we set out to evaluate and compare IPT completion rates and associated factors across DSD models. Please note that individuals in the community DSD models tend to be those who have been on ART longer because of the initial tendency to move only stable clients to community models.

Lastly, we did not conduct an implementation study but simply evaluated the new government strategy for IPT delivery that is layered within DSD models. The government of Uganda continues to layer many new innovations within DSD models; thus, evaluation of such novel ideas is paramount to guide the improvement of strategy to deliver them.

c. You didn’t include the clinical characteristics like viral load since its completion study. What about the status of completeness of data, data quality and how you had treated the data. 

Response: Thank you for this observation. Data on viral load was not extracted and hence could not be analysed; this has been added as a limitation (lines 279-281). Regarding data completeness, data quality and strategies to address, where data were missing from the electronic data system, we cross-checked with patient care cards or IPT registers and abstracted data with the help of a digital data abstraction tool installed on tablet computers (see lines 126-127). To account for missing data, multiple imputations using a multivariate normal model were used (see lines 141-142). In addition, we acknowledged and reported missing data for each variable in the tables.

d. It is senses and you have conducted regression analysis which is Poisson. Is that the recommended solution? If it is even survey, is the case to variable ratio fits the model? 

Response: Thank you for this comment. In this paper, we wanted to estimate and compare completion rates (estimated as risk ratio) across DSD models. A modified Poisson fitted by specifying robust standard errors is an appropriate and commonly used strategy, although other models like log-binomial can be used. Naimi and Whitcomb (2020) give a clear discussion and justification for use of these models; their reference has been included in the paper (lines 138 and 347).

e. All the sections of the manuscript should entail what scientifically it intends to entail. 

Response: We reviewed the manuscript and ensured that each section entails the appropriate details as guided. 

f. The Manuscript should be geared with the format of the PLOS ONE and language and statistics should be revisited and appropriate analysis and assumptions should be used and reported. 

Response: We carefully reviewed the manuscript (including the statistics) to make sure it conforms to the required format, and are happy to work with the editor further to ensure full compliance.

---

## [Editor Report · Decision Letter 3]

16 Jan 2023

PONE-D-22-21537R3Isoniazid Preventive Therapy Completion between July-September 2019: A Comparison across HIV Differentiated Service Delivery Models in UgandaPLOS ONE

Dear Dr. Mugenyi,

Thank you for submitting your manuscript to PLOS ONE. After careful consideration, we feel that it has merit but does not fully meet PLOS ONE’s publication criteria as it currently stands. Therefore, we invite you to submit a revised version of the manuscript that addresses the points raised during the review process.

We look forward to receiving your revised manuscript.

Kind regards,

Ari Samaranayaka, PhD

Academic Editor

PLOS ONE

Additional Editor Comments:

In the previous review I questioned the multiple imputation for categorical variables using multivariate normal. Authors responded saying it was done using dummy variables for all categorical variables. I believe this is impossible to do. Because all data generated by multivariate normal process for missing values are in continuous scales, not in dummy coded scales as required by categorical variables. Please include a clarification for this within the manuscript with a suitable reference.
---

## [Author Response · Author response to Decision Letter 3]

17 Jan 2023

We thank the editor for the thoughtful comment regarding our manuscript and the opportunity to revise our manuscript. We have revised the manuscript to address this comment. Below, we provide a response to the editor’s comment and reference corresponding line numbers in the text with these changes. 

a. In the previous review I questioned the multiple imputation for categorical variables using multivariate normal. Authors responded saying it was done using dummy variables for all categorical variables. I believe this is impossible to do. Because, all data generated by multivariate normal process for missing values are in continuous scales, not in dummy coded scales as required by categorical variables. Please include a clarification for this within the manuscript with a suitable reference.

Response: Thank you for this excellent observation. We have revised the results using multiple imputation chained equations (MICE) which enables us to impute missing data for both categorical and continuous variables (instead of using a multivariate normal model) (see lines 140-141). However, we noticed a very small difference in the results and in most cases the estimates remained the same (see lines 193-196, and Tables 2 and 3).

---

## [Decision Letter · Decision Letter 4]

26 Apr 2023

PONE-D-22-21537R4Isoniazid Preventive Therapy Completion between July-September 2019: A Comparison across HIV Differentiated Service Delivery Models in UgandaPLOS ONE

Dear Dr. Mugenyi,

Thank you for submitting your manuscript to PLOS ONE. After careful consideration, we feel that it has merit but does not fully meet PLOS ONE’s publication criteria as it currently stands. Therefore, we invite you to submit a revised version of the manuscript that addresses the points raised during the review process.

We look forward to receiving your revised manuscript.

Kind regards,

Dickens Otieno Onyango

Academic Editor

PLOS ONE

Journal Requirements:

Reviewers' comments:

Reviewer's Responses to Questions

**Comments to the Author**

1. If the authors have adequately addressed your comments raised in a previous round of review and you feel that this manuscript is now acceptable for publication, you may indicate that here to bypass the “Comments to the Author” section, enter your conflict of interest statement in the “Confidential to Editor” section, and submit your "Accept" recommendation.

Reviewer #1: All comments have been addressed

Reviewer #3: (No Response)

2. Is the manuscript technically sound, and do the data support the conclusions?

Reviewer #1: Partly

Reviewer #3: Yes

3. Has the statistical analysis been performed appropriately and rigorously? 

Reviewer #1: Yes

Reviewer #3: I Don't Know

4. Have the authors made all data underlying the findings in their manuscript fully available?

Reviewer #1: Yes

Reviewer #3: Yes

5. Is the manuscript presented in an intelligible fashion and written in standard English?

Reviewer #1: Yes

Reviewer #3: Yes

6. Review Comments to the Author

Reviewer #1: Review Report

1. Title of the manuscript: Organizational Creativity and Teleworking during the Spread of COVID-19 in Saudi Arabia.

2. Order of authors and co-authors: Inconsistent.

3. Scope and issues covered: Inconsistent i.e., Organizational creativity in Saudi Arabia and international agencies working in Saudi Arabia. COVID 19, according to the scholarly have already changed two long standing debates. That are Working from home and learning from home? Hence, I think teleworking is poorly described.

4. Study Period: Unspecified. Did it cover the whole period of the COVID 19 pandemic in Saudi Arabia? When was the sanction imposed and lifted? Or is the sanction continuing?

5. Methods and Materials: Is that before after study? Or after the pandemic and continuing creativity and teleworking? What is the source of those variables? Are they evaluated in Saudi Arabia? What type of tool have you used? Who collected the data? When and how? How did the quality of data were assured? You didn’t highlight the findings which had change/ Cronbach alpha’s interpretation? Why you used mean score as a measurement? What is the dependent variable exactly? Why reliability? What type of reliability (inter-reliability)? How did and when you manage it? Why validity? What type of validity/ Convergent and discriminatory/? Why not internal and external validity? Or? What is the validity and reliability adjoined status? How did you manage and when? The non-response rate is more than 15%?

6. Results, Discussions and conclusion: The tables are not self-explanatory, and you have presented all the results by table. It is short and crude. The discussion and the conclusion are not concise and logical.

7. Language and grammar:

• Incorrect use of tenses and inadequately expressing sentences.

• Short paragraphs.

• Abbreviation and acronym section is lacking.

• Use of wasted research word i.e., investigates.

• Flawed and outdated reference.

8. Statistics

Are you doing correlation or mean difference primarily?

Is there cross tabulation between organizational creativity and teleworking?

Why have you used Pearson square and did the assumptions met?

Regards,

Reviewer #3: The authors have explored a critical topic on DSD and completion rates of IPT. This area of study is important in informing efficient IPT delivery approaches among this group that has high burden of TB. The article makes an important contribution. One of the major concerns about the article is missingness in data. The variable age, had close to 50% missingness, this level of missingness is worrying and going by this, missingness is likely to have also affected other variables. The authors mention it in the limitation section but do not divulge the extent of missingness, its effect on the estimates reported and the caution readers should apply when interpreting their results. Missingness and how it was addressed should be included methods section as part of the analysis approach used. Data imputation can be done to address missingness but there are limits to imputation. This is the greatest weakness of this article.

1. The analysis evaluates IPT completion among patients who initiated IPT between July-Sep 2019, IPT is taken for 6-9 months then the outcome is evaluated. This is not a cross-sectional study. Kindly revise.

2. In the methods section the authors have a section that describes conduct of census yet this was a data review. This is confusing, kindly clarify

3. Did the investigators evaluate IPT completion rates by duration on ART? Include results.

4. I seem to have missed IPT completion rates by gender. Was this analysed?

5. One portion of the results stands out. The CCLAD unadjusted RR is not significant with a high p-value (0.996), upon adjustment this turns highly significant (p<0.001) which is not impossible but rather unusual. Review this result and confirm.

6. The assertion that “community based DSD models are more person-centered for IPT” is not accurate because the data despite showing higher completion rates does not support this statement. The authors do not have survey data neither do they present qualitative data or any other data that supports the assertion or demonstrates that the facility delivery models are less person-centered. Whereas, I fully agree with their postulation that a reduction in travel cost and time may have driven higher uptake and completion of IPT due to convenience offered by the model, this does not necessarily equate to person-centered care. Kindly revise.

7. PLOS authors have the option to publish the peer review history of their article (what does this mean?). If published, this will include your full peer review and any attached files.

Reviewer #1: No

Reviewer #3: No

---

## [Author Response · Author response to Decision Letter 4]

9 Jun 2023

Dear Editor,

We thank the Reviewers for their thoughtful comments regarding our manuscript and the opportunity to revise our manuscript further. Unfortunately, the wrong document was uploaded for Reviewer 1 so we were not able to address their concerns. However, we have revised the manuscript, which we believe has been substantially improved by incorporating the changes recommended by Reviewer 3. Below, we provide a point-by-point response to the Reviewer’s comments and reference corresponding line numbers in the text with these changes. 

Reviewer #3: 

The authors have explored a critical topic on DSD and completion rates of IPT. This area of study is important in informing efficient IPT delivery approaches among this group that has high burden of TB. The article makes an important contribution. One of the major concerns about the article is missingness in data. The variable age, had close to 50% missingness, this level of missingness is worrying and going by this, missingness is likely to have also affected other variables. The authors mention it in the limitation section but do not divulge the extent of missingness, its effect on the estimates reported and the caution readers should apply when interpreting their results. Missingness and how it was addressed should be included methods section as part of the analysis approach used. Data imputation can be done to address missingness but there are limits to imputation. This is the greatest weakness of this article.

Response: Thanks so much for raising this very important issue. We agree that having a high proportion of missing data in the Age at IPT initiation variable, could have affected the results even with multiple imputation. We noted that we had not produced results for complete case analysis which was an error and could have prevented one from observing the effect of missing data. We updated the results to include complete case analysis and are now considering the results from the multiple imputation as sensitivity analysis. We observe that adjusted results from multiple imputation generally did not differ from those from complete case analysis when looking at factors associated with IPT completion in each DSD model (Table 3). However, when comparing IPT completion across DSD models (Table 2), there is an association with CCLAD and CDDP in the complete case analysis, but the effect disappears with multiple imputation. When analyzing the missing age with DSD model, we noted that all participants in CCLAD had data available on age, which could explain the difference in observed results. Revisions have been made to the manuscript to clarify the effect of missing data. See lines 143-150; Lines 187-192; Table 2; and Table 3.

1. The analysis evaluates IPT completion among patients who initiated IPT between July-Sep 2019, IPT is taken for 6-9 months then the outcome is evaluated. This is not a cross-sectional study. Kindly revise.

Response: Thank you for this observation. We have revised the study design as follows “This was a retrospective study with review of electronic medical records and patient registers for PLHIV who initiated IPT during the period of July-September 2019.” (Lines 110-111)

2. In the methods section the authors have a section that describes conduct of census yet this was a data review. This is confusing, kindly clarify

Response: This has been clarified as data review by revising the statement to read as follows “We conducted a medical records review of all PLHIV who had initiated IPT at the three selected health facilities during July-September 2019.” (Line 120-121)

3. Did the investigators evaluate IPT completion rates by duration on ART? Include results.

Response: Yes, we investigated the effect of duration on ART at time of ITP initiation. However, this was not significantly associated with IPT completion in all DSD models except CCLAD where it’s only significant in unadjusted analysis. After adjusting for other factors, it became insignificant both for complete case analysis and with multiple imputation. Results are now shown in Table 3 for the CCLAD model.

4. I seem to have missed IPT completion rates by gender. Was this analysed?

Response: Thank you for this observation. We have now added analysis of completion rate by gender in each DSD model, and its effect shows up in FBG and CCLAD models. Results are now presented in Table 3 and in lines 204-205.

5. One portion of the results stands out. The CCLAD unadjusted RR is not significant with a high p-value (0.996), upon adjustment this turns highly significant (p<0.001) which is not impossible but rather unusual. Review this result and confirm.

Response: Thank you for catching this. We revisited the multiple imputation method using multiple imputation chained equations (MICE) on factor variables instead of using the multivariate normal models on dummy variables (derived from the factor variables) which we had used previously. With the MICE approach, the p-values for unadjusted and multiple imputation results are now consistent (see Table 2). 

6. The assertion that “community based DSD models are more person-centered for IPT” is not accurate because the data despite showing higher completion rates does not support this statement. The authors do not have survey data neither do they present qualitative data or any other data that supports the assertion or demonstrates that the facility delivery models are less person-centered. Whereas, I fully agree with their postulation that a reduction in travel cost and time may have driven higher uptake and completion of IPT due to convenience offered by the model, this does not necessarily equate to person-centered care. Kindly revise.

Response: Thank you for noting that; the statement has been deleted from the Discussion section.

---

## [Decision Letter · Decision Letter 5]

7 Jul 2023

PONE-D-22-21537R5Isoniazid Preventive Therapy Completion between July-September 2019: A Comparison across HIV Differentiated Service Delivery Models in UgandaPLOS ONE

Dear Dr. Mugenyi,

Thank you for submitting your manuscript to PLOS ONE. After careful consideration, we feel that it has merit but does not fully meet PLOS ONE’s publication criteria as it currently stands. Therefore, we invite you to submit a revised version of the manuscript that addresses the points raised during the review process.

We look forward to receiving your revised manuscript.

Kind regards,

Dickens Otieno Onyango

Academic Editor

PLOS ONE

Journal Requirements:

Reviewers' comments:

Reviewer's Responses to Questions

**Comments to the Author**

1. If the authors have adequately addressed your comments raised in a previous round of review and you feel that this manuscript is now acceptable for publication, you may indicate that here to bypass the “Comments to the Author” section, enter your conflict of interest statement in the “Confidential to Editor” section, and submit your "Accept" recommendation.

Reviewer #4: (No Response)

Reviewer #5: (No Response)

2. Is the manuscript technically sound, and do the data support the conclusions?

Reviewer #4: Partly

Reviewer #5: Yes

3. Has the statistical analysis been performed appropriately and rigorously? 

Reviewer #4: Yes

Reviewer #5: Yes

4. Have the authors made all data underlying the findings in their manuscript fully available?

Reviewer #4: Yes

Reviewer #5: Yes

5. Is the manuscript presented in an intelligible fashion and written in standard English?

Reviewer #4: Yes

Reviewer #5: Yes

6. Review Comments to the Author

Reviewer #4: • Abstract line 56-58: How were clients allocated to various DSD models. Could client allocation have influenced selection of patients with higher probability of getting certain TPT outcomes? Are there possible limitations of the study kindly include these in the abstract.

• Line 66-67: Good information on ART coverage- however you need to quantify this as a percege relative to the need to give the reader a sense of how high ART coverage as you highlight TB as a major problem among PLHIV

• Line 118 need to highlight what proportion of the clients were excluded based on missing data on ART initiation

• Line 136-138- was the Poisson model adjusted for ART treatment adherence? This could be a potential source of bias in allocation to different DSD models with patients in facility models likely to less adherent to TPT.

• Line 157-164, consider being consistent in reporting your results. Include numerator and percentage for all proportions we note that this is provided for some and not for others.

• Check for punctuations and spacing across the document

• Line 161, 27.6% does not constitute the majority but rather the largest proportion. You may want to think through this.

• Line 190: What about adjustment for ART adherence? This needs to be included in the model as it is a potential source of bias

• Line 285-292: One major potential source of bias that needs to be highlighted in this study is the fact that in HIV programs patients who are stable on ART and more likely to adhere to HIV treatment and TPT are selected into community DSD models while those who are unstable and less adherent to treatment receive care in facility-based models to ensure that they benefit from additional health care worker support. Thus, the poorer outcomes in facility DSD models may be attributable to selection bias

• Line 126 We also need to have a clear determination of how TPT completion was defined. How did the study team verify TPT completion in community-based DSD models given that TPT was dispensed for longer periods? Was dispensing of the last doses of TPT considered completion (as stated in line 126?). What are chances that some clients on DSD were misclassified as having completed TPT when indeed they did not do so? Include in your manuscript how completion was verified

Reviewer #5: The manuscript explores IPT completion rates (and related factors) in light of various DSD models in eastern Uganda. The overall sample size is adequate though with significant missing variables – to which the authors try to address. Some sub-analyses also end up being carried out on fewer numbers due to the many variable points. As it has gone through a round of review process, I would just point out a few areas to further improve the manuscript

1. Line 68: one in three has hyphens in between

2. Lines 75-6: the sentence IPT is ‘currently’ the most common form of TPT in resource-limited settings may be intriguing at the time of publication. Shorter TPT regimens are increasingly being used even in resource-limited settings. Thus, either back this sentence with a reference or change tense to reflect past or let it refer to the bulk of PLHIV already in care.

3. Line 85: change ‘recommend’ to ‘recommended’, as the 2014 Uganda IPT guidelines may have been surpassed by revisions.

4. Lines 137-8: The poisson model estimated the RRs by adjusting for multiple variables. Was collinearity considered between ART duration, treatment line and regimen? Was it tested and adjusted appropriately? E.g., the longer one is on ART the likelihood of moving to 2nd and 3rd treatment lines. Equally, guidelines indicate that first lines be DTG-based (an INSTI) leading to potential relationship between treatment line and regimen.

5. Under tables 1 and 3, include the DSD acronyms in full.

6. Line 219 in discussion, also in results table 3 - IPT completion at facility-based models differed between sites: What could have been the likely reason of the consistently lower IPT completion in TASO Soroti facility models? Even an intelligent speculation would suffice in the discussion.

7. Line 296: Delete the word ‘concrete’ in …provide concrete evidence…this is superfluous in this context.

7. PLOS authors have the option to publish the peer review history of their article (what does this mean?). If published, this will include your full peer review and any attached files.

Reviewer #4: No

Reviewer #5: **Yes: **Dr Philip Owiti

---

## [Author Response · Author response to Decision Letter 5]

31 Jul 2023

Dear Editor,

We thank the Reviewers for their thoughtful comments regarding our manuscript and the opportunity to revise our manuscript further. Below, we provide a point-by-point response to the Reviewer’s comments and reference corresponding line numbers in the text with these changes. 

Reviewer #4: 

Abstract line 56-58: How were clients allocated to various DSD models. Could client allocation have influenced selection of patients with higher probability of getting certain TPT outcomes? Are there possible limitations of the study kindly include these in the abstract.

Response: Thank you for your comment. Stable clients, who are virally suppressed are allocated to three models namely, CCLAD, CDDP and FTDR while unstable clients (virally not suppressed) are allocated to FBIM and FBG. Additional clarity on how patients were allocated to these different models was added to the Methods section (lines 83-85). 

We agree that patient allocation could have influenced higher IPT completion rates in certain DSD models and added the following sentence to the discussion “It is possible that the allocation of stable clients to community-based models could explain the higher rates of IPT completion among clients in these DSD models” (Lines 259-260). We also added this as a potential limitation (lines 304-307).

Study limitations are outlined in the Discussion section (Lines 302-311). However, due to the limited word count for the abstract, we could not include these limitations in the abstract.

• Line 66-67: Good information on ART coverage- however you need to quantify this as a percege relative to the need to give the reader a sense of how high ART coverage as you highlight TB as a major problem among PLHIV

Response: Thank you for pointing this out. We have revised as guided as seen in the quote “Despite the impressive scale-up of HIV treatment, with 61.5% (23.3/37.9 million) people living with HIV (PLHIV) in 2018 receiving antiretroviral therapy (ART) at the end of 2018, TB remains the leading cause of death among people living with HIV (PLHIV), accounting for approximately one-in-three AIDS-related deaths” (Lines 66-69)

• Line 118 need to highlight what proportion of the clients were excluded based on missing data on ART initiation

Response. Thank you for this comment. A proportion with missing data on ART initiation has been added (line 119). 

• Line 136-138- was the Poisson model adjusted for ART treatment adherence? This could be a potential source of bias in allocation to different DSD models with patients in facility models likely to less adherent to TPT.

Response. Thank you very much for this important observation. Indeed, we would expect patients with better adherence to ART to also have better adherence to IPT, and hence potentially higher IPT completion rate. Unfortunately, data on ART adherence was not readily available and so was not extracted. This has been added to the limitations (Line 308).

• Line 157-164, consider being consistent in reporting your results. Include numerator and percentage for all proportions we note that this is provided for some and not for others.

Response. Thank you! We have now included both numbers (numerators) and percentages as appropriate (Lines 159-168).

• Check for punctuations and spacing across the document

Response: This has been addressed, thank you! 

• Line 161, 27.6% does not constitute the majority but rather the largest proportion. You may want to think through this.

Response: Thank you for your suggestion. This has been addressed by using largest proportion instead of majority (line 164).

• Line 190: What about adjustment for ART adherence? This needs to be included in the model as it is a potential source of bias

Response: Again, thank you for this observation. Like mentioned above, we did not extract data on ART adherence and this is pointed out in the limitations (line 308).

• Line 285-292: One major potential source of bias that needs to be highlighted in this study is the fact that in HIV programs patients who are stable on ART and more likely to adhere to HIV treatment and TPT are selected into community DSD models while those who are unstable and less adherent to treatment receive care in facility-based models to ensure that they benefit from additional health care worker support. Thus, the poorer outcomes in facility DSD models may be attributable to selection bias.

Response: We agree with your observation and suggestion. We have included this as a limitation due to potential selection bias (Lines 304-305).

• Line 126 We also need to have a clear determination of how TPT completion was defined. How did the study team verify TPT completion in community-based DSD models given that TPT was dispensed for longer periods? Was dispensing of the last doses of TPT considered completion (as stated in line 126?). What are chances that some clients on DSD were misclassified as having completed TPT when indeed they did not do so? Include in your manuscript how completion was verified

Response: This study was a retrospective review of clients’ medical records and therefore we were not able to verify the IPT completion status. The following statement has been included among the limitations “Fourth, we could not verify IPT completion and we relied on the medical record systems where the completion status was documented.” (Lines 310-311).

 

Reviewer #5: 

The manuscript explores IPT completion rates (and related factors) in light of various DSD models in eastern Uganda. The overall sample size is adequate though with significant missing variables – to which the authors try to address. Some sub-analyses also end up being carried out on fewer numbers due to the many variable points. As it has gone through a round of review process, I would just point out a few areas to further improve the manuscript

1. Line 68: one in three has hyphens in between

Response: Done, thank you! (Line 68).

2. Lines 75-6: the sentence IPT is ‘currently’ the most common form of TPT in resource-limited settings may be intriguing at the time of publication. Shorter TPT regimens are increasingly being used even in resource-limited settings. Thus, either back this sentence with a reference or change tense to reflect past or let it refer to the bulk of PLHIV already in care.

Response: Agreed with the reviewer. We dropped “currently” and revised the sentence to “Isoniazid preventive therapy (IPT), which involves daily administration of isoniazid and vitamin B6 for six months, was the most common TPT in resource-limited settings at the time of this study” (Lines 74-76).

3. Line 85: change ‘recommend’ to ‘recommended’, as the 2014 Uganda IPT guidelines may have been surpassed by revisions.

Response: Done, thank you! (Line 86).

4. Lines 137-8: The poisson model estimated the RRs by adjusting for multiple variables. Was collinearity considered between ART duration, treatment line and regimen? Was it tested and adjusted appropriately? E.g., the longer one is on ART the likelihood of moving to 2nd and 3rd treatment lines. Equally, guidelines indicate that first lines be DTG-based (an INSTI) leading to potential relationship between treatment line and regimen.

Response: Thank you for pointing out this oversight of not reporting multicollinearity tests in the manuscript. We used variance inflation factor (VIF) to test for multicollinearity. For all our final fits for which we report adjusted estimates, VIF values were less than 3 indicating no issues for multicollinearity. This has been clarified by adding the following sentence under the data analysis section “Variance inflation factor (VIF) was used to test for multicollinearity in independent variables (Line 145). Also, results on multicollinearity have been added to the results section (Lines 183-184 and Lines 196-197).

5. Under tables 1 and 3, include the DSD acronyms in full.

Response: Done, thank you! (Tables 1 and 3).

6. Line 219 in discussion, also in results table 3 - IPT completion at facility-based models differed between sites: What could have been the likely reason of the consistently lower IPT completion in TASO Soroti facility models? Even an intelligent speculation would suffice in the discussion.

Response: Around the study time, TASO Soroti experienced episodes of stock-out of IPT yet several clients were already initiated on it. This, partly explain the comparatively lower IPT completion rate, especially among individuals in the facility models. By design, unlike those in the community model, PLHIV at facility-based model receive only one to two months of ART and IPT refills due to their being unstable on ART. Therefore, during this period of frequent stock-out of IPT, PLHIV at TASO Soroti facility were disadvantaged whereby several of them could not get IPT drug refills. Secondly, the poorer IPT completion rate at TASO Soroti could have been due to the fact that PLHIV at facility comprise of the most unstable groups with poorer adherence to ART. A paragraph has been added to the discussions (Lines 277-286)

7. Line 296: Delete the word ‘concrete’ in …provide concrete evidence…this is superfluous in this context.

Response: Done, thank you! (Line 318).

---

## [Decision Letter · Decision Letter 6]

13 Oct 2023

PONE-D-22-21537R6Isoniazid Preventive Therapy Completion between July-September 2019: A Comparison across HIV Differentiated Service Delivery Models in UgandaPLOS ONE

Dear Dr. Mugenyi,

Thank you for submitting your manuscript to PLOS ONE. After careful consideration, we feel that it has merit but does not fully meet PLOS ONE’s publication criteria as it currently stands. Therefore, we invite you to submit a revised version of the manuscript that addresses the points raised during the review process.

We look forward to receiving your revised manuscript.

Kind regards,

Dickens Otieno Onyango

Academic Editor

PLOS ONE

Journal Requirements:

Reviewers' comments:

Reviewer's Responses to Questions

**Comments to the Author**

1. If the authors have adequately addressed your comments raised in a previous round of review and you feel that this manuscript is now acceptable for publication, you may indicate that here to bypass the “Comments to the Author” section, enter your conflict of interest statement in the “Confidential to Editor” section, and submit your "Accept" recommendation.

Reviewer #6: (No Response)

Reviewer #7: (No Response)

2. Is the manuscript technically sound, and do the data support the conclusions?

Reviewer #6: Yes

Reviewer #7: Partly

3. Has the statistical analysis been performed appropriately and rigorously? 

Reviewer #6: Yes

Reviewer #7: I Don't Know

4. Have the authors made all data underlying the findings in their manuscript fully available?

Reviewer #6: Yes

Reviewer #7: Yes

5. Is the manuscript presented in an intelligible fashion and written in standard English?

Reviewer #6: Yes

Reviewer #7: Yes

6. Review Comments to the Author

Reviewer #6: Introduction

Line 85.

Kindly quote the 2018 updated guidelines on TB preventive therapy which were in use at the time this study was carried out

Move the description about differentiated service delivery models to the study setting section.

Methods

Study Setting and Population

The study setting and population is inadequately described

The description about differentiated service delivery models and patients eligible for each should be under the study setting not in the introduction

Authors should then describe the criteria-if any for initiating patients on IPT, the number of clinic follow-up visits conducted while on IPT-if any, the number of months of IPT dispensed at each clinic refill visit for each DSD model.

Study Outcome: How was the main study outcome “TPT completion” measured? Was it the same across all study groups?

Results

Authors should present only the results. Text explaining the results e.g., Lines 189-191 should be moved to the discussion.

Lines 184 and 196, read “there were no issues of collinearity”. Authors should consider changing to ….”there was no collinearity.”

Line 198-199 read….. “results from the complete case analysis are similar to those from multiple imputation, which implies that the latter method did not produce biased estimates”. Did the did the authors mean the former method? i.e. that the complete case analysis did not produce biased estimates?

Authors should consider including some results on common reasons for IPT non- completion since this data was collected – according to lines 128 and 129 in the data collection section

In the discussion section, the authors explained that IPT completion was lower in facility-based models due to episodes of drug stock outs. Was a sensitivity analysis conducted exclude patients affected by the stock out period to check if treatment completion at health facilities still remained lower? i.e. to check that the authors are really measuring IPT adherence not IPT drug availability?

Discussion

No comments

Reviewer #7: Dear authors: I am reviewing this manuscript for the first time and I appreciate the changes made based on the first round of revisions but I think there are a few more items to revise before publication:

The IPT completion is very very high across all models. This systematic review PMID 34492003 and this other one PMID 27522233 find much lower proportions. How is it so high in your study setting, even for facility based delivery models? Please provide some additional information to understand this in the context of other studies. This should be explained in the study context and in the discussion.

Do we know how many were eligible and thus can include the completion out of the elegible and not only among those who started?

I missed the definition of IPT completion in the methods. Was it by number of doses picked up? Was there a range?

Figure 2 should include confidence intervals and, if used, other statistical tests to compare the completion among hte models.

Please reduce the use of acronyms as it makes it difficult to read. I suggest to keep only the very known ones (HIV, ARV, IPT) but not for each model, each study site, etc.

7. PLOS authors have the option to publish the peer review history of their article (what does this mean?). If published, this will include your full peer review and any attached files.

Reviewer #6: No

Reviewer #7: No

---

## [Author Response · Author response to Decision Letter 6]

9 Nov 2023

Dear Editor,

We thank the Reviewers for their thoughtful comments regarding our manuscript and the opportunity to revise our manuscript further. We have revised the manuscript to respond to comments from Reviewers #6 and #7, and we believe it been substantially improved by incorporating the changes recommended by these Reviewers. Below, we provide a point-by-point response to the Reviewers’ comments and reference corresponding line numbers in the text with these changes. 

Reviewer #6: 

Introduction

Line 85.

Kindly quote the 2018 updated guidelines on TB preventive therapy which were in use at the time this study was carried out.

Response: Thank you for this suggestion. Revision is made to quote the 2018 guidelines. Line 111.

Move the description about differentiated service delivery models to the study setting section.

Response: The description about differentiated service delivery models has been moved to study setting section as guided. Lines106-116. 

Methods

Study Setting and Population

The study setting and population is inadequately described

The description about differentiated service delivery models and patients eligible for each should be under the study setting not in the introduction

Response: The study setting and population section has been extended by transferring the description about differentiated service delivery models into this section as guided. Lines106-116

Authors should then describe the criteria-if any for initiating patients on IPT, the number of clinic follow-up visits conducted while on IPT-if any, the number of months of IPT dispensed at each clinic refill visit for each DSD model.

Response: Thank you for pointing these out. We have included the following in our revised manuscript. “To be eligible for IPT, a client must be living with HIV and be on ART for any duration. Those who had active pulmonary TB disease were not eligible for IPT initiation. The number of clinic follow-up visits and months of IPT dispensed at each visit varied by DSD models. Clinically stable clients in the community-based DSD models received multi-month (3 or more months) of synchronized IPT and ART refill per visit, while clinically unstable clients in the facility-based DSD models received 1-2 months of synchronized IPT and ART refills per visit. To complete TPT, clients in the community-based DSD models required only two clinic visits, while those in the facility-based models required 3-6 clinic visits within the 6-9 months of IPT initiation”. Lines 119-126.

Study Outcome: How was the main study outcome “TPT completion” measured? Was it the same across all study groups?

Response: TPT completion was measured as a proportion of PLHIV who completed the course of isoniazid treatment within 6-9 months among those who initiated TPT. We have clarified the definition of TPT completion as shown in lines 142-143.

Results

Authors should present only the results. Text explaining the results e.g., Lines 189-191 should be moved to the discussion.

Response: The text explaining results has been transferred to the discussion. Lines 318-319.

Lines 184 and 196, read “there were no issues of collinearity”. Authors should consider changing to ….”there was no collinearity.”

Response: This has been revised as guided. Lines 196 & 206

Line 198-199 read….. “results from the complete case analysis are similar to those from multiple imputation, which implies that the latter method did not produce biased estimates”. Did the did the authors mean the former method? i.e. that the complete case analysis did not produce biased estimates?

Response: Thanks, yes, the statement refers to complete case analysis, the former method and this has been corrected. Line 208.

Authors should consider including some results on common reasons for IPT non- completion since this data was collected – according to lines 128 and 129 in the data collection section

Response: Results on reasons for not completing IPT are now provided. Lines 184-186.

In the discussion section, the authors explained that IPT completion was lower in facility-based models due to episodes of drug stock outs. Was a sensitivity analysis conducted exclude patients affected by the stock out period to check if treatment completion at health facilities still remained lower? i.e. to check that the authors are really measuring IPT adherence not IPT drug availability?

Response: We acknowledge that a sensitivity analysis of drug stock out would help improve our results on IPT completion. However, note that we only obtained qualitative data on stock out and so the sensitivity analysis could not be performed. Secondly, we added further detail write up reflecting the fact that the stock out affected everyone but differentially impacted those in the facility-based designs due to the inherent designs of DSD models which enabled PLHIV in the community-based DSD models to require only 1-2 IPT refills to complete their TPT doses while those in the facility-based model required 3-6 IPT refills. 

Reviewer #7: 

Dear authors: I am reviewing this manuscript for the first time and I appreciate the changes made based on the first round of revisions but I think there are a few more items to revise before publication:

Response: Thank you for taking time to review our manuscript and for your valuable comments.

The IPT completion is very very high across all models. This systematic review PMID 34492003 and this other one PMID 27522233 find much lower proportions. How is it so high in your study setting, even for facility based delivery models? Please provide some additional information to understand this in the context of other studies. This should be explained in the study context and in the discussion.

Response: Thank you so much for these very useful citations on latent TB cascade of care outcomes. We have used these citations to contextualize our findings and enrich our discussion. Lines 251-254

Do we know how many were eligible and thus can include the completion out of the eligible and not only among those who started?

Response: Thank you for your thoughts on estimating completion out of those eligible for IPT. However, the aim of this paper was to estimate completion among those that initiated IPT and thus we restricted data extraction to those who initiated IPT and not those who were eligible for IPT. 

I missed the definition of IPT completion in the methods. Was it by number of doses picked up? Was there a range?

Response: IPT completion was measured as a proportion of PLHIV who completed the course of IPT within 6-9 months among those who initiated IPT. This has been clarified in the paper. Lines 142-143

Figure 2 should include confidence intervals and, if used, other statistical tests to compare the completion among hte models.

Response: Figure 2 has been edited to include confidence intervals and the test statistic used. 

Please reduce the use of acronyms as it makes it difficult to read. I suggest to keep only the very known ones (HIV, ARV, IPT) but not for each model, each study site, etc.

Response: We have minimized abbreviations as much as possible. We now state site names in full everywhere. However, because the types of DSD models are frequently mentioned in this paper including in the abstract, are well explained in Figure 1, and given in the list of abbreviations section, we have left these abbreviations in the manuscript.

---

## [Decision Letter · Decision Letter 7]

10 Dec 2023

Isoniazid Preventive Therapy Completion between July-September 2019: A Comparison across HIV Differentiated Service Delivery Models in Uganda

PONE-D-22-21537R7

Dear Dr. Mugenyi,

We’re pleased to inform you that your manuscript has been judged scientifically suitable for publication and will be formally accepted for publication once it meets all outstanding technical requirements.

Kind regards,

Dickens Otieno Onyango

Academic Editor

PLOS ONE

Additional Editor Comments (optional):

Reviewers' comments:

Reviewer's Responses to Questions

**Comments to the Author**

1. If the authors have adequately addressed your comments raised in a previous round of review and you feel that this manuscript is now acceptable for publication, you may indicate that here to bypass the “Comments to the Author” section, enter your conflict of interest statement in the “Confidential to Editor” section, and submit your "Accept" recommendation.

Reviewer #6: All comments have been addressed

Reviewer #7: All comments have been addressed

2. Is the manuscript technically sound, and do the data support the conclusions?

Reviewer #6: Yes

Reviewer #7: Yes

3. Has the statistical analysis been performed appropriately and rigorously? 

Reviewer #6: Yes

Reviewer #7: Yes

4. Have the authors made all data underlying the findings in their manuscript fully available?

Reviewer #6: Yes

Reviewer #7: Yes

5. Is the manuscript presented in an intelligible fashion and written in standard English?

Reviewer #6: (No Response)

Reviewer #7: Yes

6. Review Comments to the Author

Reviewer #6: All comments have been addressed fully except one.

The definition of TPT completion is stated by the authors as "a proportion of PLHIV who completed the course of isoniazid treatment within 6-9 months among those who initiated TPT." However, the word completion should not be used to define completion.

The authors should consider, defining TPT completion as the proportion of PLHIV who received/picked up all required doses of isoniazid from the health facility / community distribution point within 6-9 months of initiating TPT.

Reviewer #7: Thanks for addressing all comments, the manuscript looks good and I have no further comments.

7. PLOS authors have the option to publish the peer review history of their article (what does this mean?). If published, this will include your full peer review and any attached files.

Reviewer #6: No

Reviewer #7: No

---

## [Editor Report · Acceptance letter]

20 Dec 2023

PONE-D-22-21537R7 

PLOS ONE

Dear Dr. Mugenyi, 

I'm pleased to inform you that your manuscript has been deemed suitable for publication in PLOS ONE. Congratulations! Your manuscript is now being handed over to our production team.

Kind regards, 

on behalf of

Dr. Dickens Otieno Onyango 

Academic Editor

PLOS ONE